# APTBench: Benchmarking Agentic Potential of Base LLMs During Pre-Training

## Abstract

With the rapid development of LLM-based agents, there is a growing trend to incorporate agent-specific data into the pre-training stage of LLMs, aiming to better align LLMs with real-world autonomous task execution. However, current pre-training benchmarks primarily focus on *isolated and static skills*, *e.g.*, common knowledge or mathematical/code reasoning, and fail to reflect model's agentic capabilities. On the other hand, agent benchmarks are typically designed for post-trained models, requiring *multi-turn task execution abilities* that base models struggle to support. Thus, there is a compelling need for a benchmark that can evaluate **agentic potentials** during pre-training and guide the model training more effectively. To address this gap, we propose **APTBench**, a framework that converts real-world agent tasks and successful trajectories into *multiple-choice or text completion questions* tailored for base models. It focuses on core agentic abilities, *e.g.*, planning and action, and covers key agent scenarios, *software engineering* and *deep research*. Compared to existing general-purpose benchmarks, APTBench offers a more predictive signal of a model's downstream performance as an agent, while remaining significantly more lightweight and cost-effective than full-scale, end-to-end agent evaluations after post-training.

## 1 Introduction

LLM-based agents have recently demonstrated remarkable proficiency in real-world applications, *e.g.*, Claude Code Agent (Claude, 2025) and Deep Research of OpenAI (OpenAI, 2025). As a result, increasing attention is being given to enhancing agent performance for practical use, often through post-training techniques like instruction fine-tuning and agentic reinforcement learning (Wang et al., 2025; Du et al., 2025b; Xi et al., 2025a; Wang et al., 2024a). Recent studies have shown that enhancing agentic capabilities during pre-training[1] can improve their performance on downstream agent tasks (Wu et al., 2025; Zeng et al., 2025; Su et al., 2025), as it is widely recognized that a model's core capabilities are established during pre-training (Yue et al., 2025). Although the base model serves as the foundation for the downstream agentic capabilities, **there remains a lack of proper measures to quantify these agentic potentials during pre-training**.

Developing an agent-oriented benchmark for the pre-training stage enables us to *monitor and guide the agentic potentials of a model from an early stage*. This is essential for the expensive pre-training runs, because altering a model's core competencies after pre-training is not only prohibitively expensive but also frequently yields suboptimal outcomes. By contrast, assessing these capabilities during pre-training allows researchers to make informed decisions about the training data mix or model architecture designs at a foundational stage. Therefore, the development of robust methods for evaluating a base model's potential for agent-based tasks is of critical importance.

**The general benchmarks for pre-training stage fails to reflect model's agentic potential** as they are disconnected from real-world agent applications and capabilities. Most existing benchmarks, *e.g.*, MATH (Hendrycks et al., 2021), GSM8K (Cobbe et al., 2021), EvalPlus (Liu et al., 2023a), GPQA (Rein et al., 2024), MMLU (Hendrycks et al., 2020), are static and single-turn, designed only to assess a model's isolated skills, such as knowledge, mathematical reasoning, or code generation. In contrast, real-world applications demand an agent make dynamic decisions based on external

---

[1]For simplicity, we use pre-training to refer to the stages before post-training, which yields the base model.

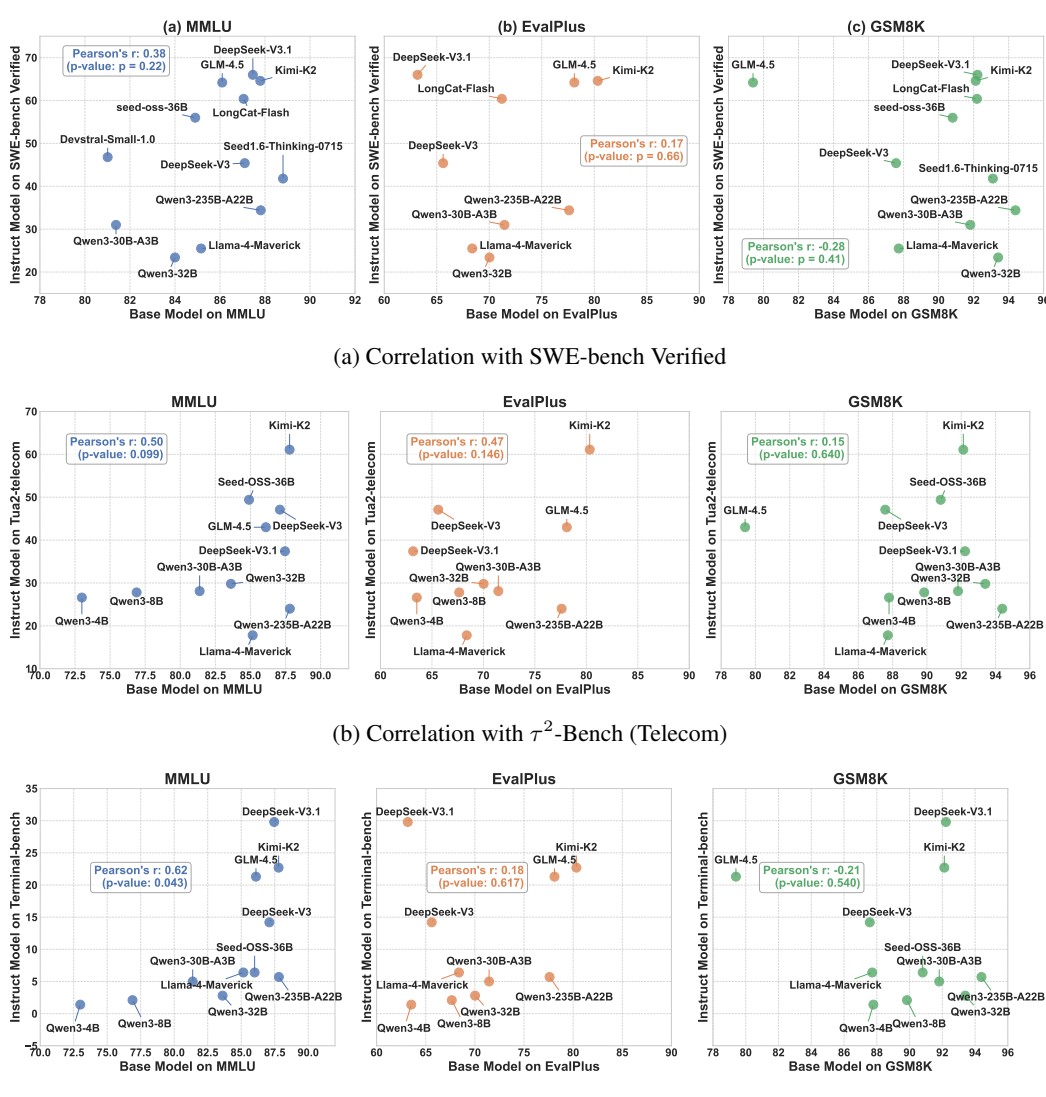

Figure 1: The correlation between model's performance on general benchmarks (MMLU, EvalPlus, and GSM8K) and various agent benchmarks. **(a)** Comparison with SWE-bench Verified. As noted, six models with similar MMLU scores (86-88) show a 30-point difference on SWE-Bench, indicating weak correlation. **(b)** Comparison with $\tau^2$-Bench. **(c)** Comparison with Terminal Bench. Across all figures, low r-values and high p-values suggest that general capabilities do not strictly guarantee agentic performance.

feedback and perform multi-turn interactions. The current general benchmarks cannot directly measure this "planning-action-feedback" loop, and therefore fail to reflect a model's performance when facing dynamic and uncertain environments.

As shown in Figure 1, we present the performance of several *base models* on representative general benchmarks (MMLU for knowledge, EvalPlus for coding, and GSM8K for math), as well as the results of their *instruct versions* on a widely recognized agent benchmark, SWE-bench Verified (Jimenez et al., 2023). It is evident that results of these general benchmarks show a **weak correlation** with the performance on the downstream agent task, while more results and analysis can be found in Appendix B. Additionally, the scores of the models on these general benchmarks do not differ significantly, showing limited distinction, whereas this is not the case for agent tasks. Thus, current general benchmarks for base models are inadequate for measuring their agent capabilities.

Although evaluating base models for agent capabilities offers much benefits, **it is not feasible to evaluate them on real-world, multi-turn agent tasks in an end-to-end manner**. As the base models have not yet undergone post-training, they struggle with complex instructions and multi-turn tasks. However, the current agent benchmarks is mainly for instruct models (Jimenez et al., 2023; Barres et al., 2025; TTB-Team, 2025), making them not suitable for base model evaluation.

To address this gap, we propose **APTBench**, the first benchmark designed to evaluate the **A**gent **Po**Tential of base models. *Firstly*, we introduce a general framework for benchmark construction, which transforms real-world agent tasks and successful trajectories into *multiple-choice or text completion questions* suitable for base models. This can bypass the lack of instruction-following capabilities for base models. These questions are specifically designed to assess core agentic capabilities in multi-turn interactions, *i.e.*, *planning, action, and domain-specific atomic abilities*. **Next**, we apply this framework to create challenging benchmarks on two critical scenarios, *software engineering* (SWE) and *deep research* (DR). For each domain, we cover representative tasks, *e.g.*, open-ended and close-ended questions for deep research, environment setup and issue fixing for software engineering, and design specific question types to assess the model's core agent capabilities.

This benchmark provides the *first* feasible solution for evaluating the agent potentials of base models and offers quantitative performance metrics that can guide the agentic pre-training. Our construction method is easy to extend to other agent scenarios. The extensive experiments across small, medium, and large models show that APTBench is closely correlated with final agentic capabilities.

## 2 BENCHMARKING AGENT POTENTIALS OF BASE MODELS

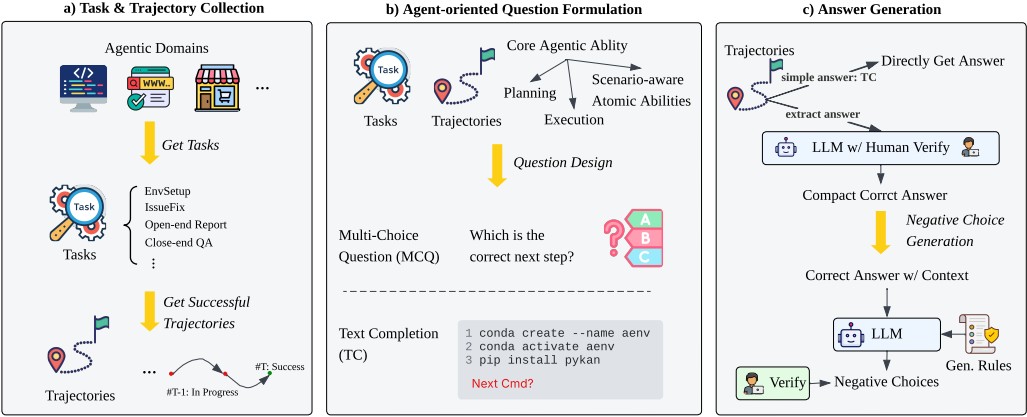

Figure 2: The construction process of APTBench. Firstly, we collect agentic tasks and successful trajectories from real-world domains. Then, we generate multi-choice question and text completion tasks through correct answer extraction and negative choices generation processes.

### 2.1 CONSTRUCTION PRINCIPLE

To address the challenge of base models being unable to execute agent tasks, we convert complex, multi-turn agent tasks and their trajectories in real-world scenarios into multiple-choice or text completion questions suitable for evaluating base models, which consists of the following key steps:

**Task and Trajectory Collection.** We first select a scenario aligned with real-world agent applications, *e.g.*, software engineering. Then, we gather relevant tasks of the application and collect interaction trajectories generated by humans or agents. We only record the successful attempt for human trajectories. The agent trajectories, which are in the form of Plan-Action-Feedback, undergo rejection sampling and human validation to ensure they successfully solve the task.

**Agent-oriented Question Formulation.** We identify several core abilities that agents must demonstrate during their interactions, including

- **Planning:** This includes overall planning for high-level task decomposition & organization, and stepwise planning, dynamic plan adjusting based on external feedback.
- **Action:** The ability to correctly complete the next action based on the task and existing trajectory, *e.g.*, tool invocation or conclusion generation.
- **Atomic Abilities:** Essential, scenario-specific abilities that are tightly linked to the task at hand, such as citation generation in deep research or bug location in software engineering.

Next, we extract content related to each ability from these agent tasks and trajectories and transform it into next-token prediction questions (in multiple-choice or text completion format). For example, when assessing stepwise planning abilities, we provide the task descriptions and the first $T$-th steps of a trajectory, and require the model to choose the plan for next step.

**Answer Generation.** Typically, we use the original overall plan or stepwise plan/action at the $T + 1$ step from the trajectory as the correct answer. For longer or more ambiguous questions, *e.g.*, situations where there may be multiple valid plans or actions, we convert them into multiple-choice questions which has a definite best answer. We then use LLMs to degrade the correct answer and generate incorrect choices, ensuring that the correct answer is always the optimal one. For concise and clear answers, such as single-line command execution, we adapt the text completion format. Both question types undergo human validation to ensure their accuracy.

Following the above principles, we build benchmarks for software engineering and deep research. The construction approach is general and can be easily extended to other agent scenarios.

Table 1: Different tasks of APTBench-SWE (Software Engineering) and APTBench-DR (Deep Research). MCQ denotes Multi-Choice Question and TC is text completion task.

| Scenario | Task | Ability | Description | Format | Length | Size |
|---|---|---|---|---|---|---|
| SWE | EnvSetup | Planning | Select the best overall setup plan based on requirements | MCQ | 16-32K | 437 |
| | | Action | Write next command based on existing trajectory | TC | <4K | 1084 |
| | | Atomic | Select solution for errors during setup (Handle_Error) | MCQ | 4-16K | 147 |
| | IssueFix | Planning | Select the best next stepwise fixing plan based on issue | MCQ | 32-128K | 243 |
| | | Action | Write next command based on existing trajectory | TC | 32-128K | 241 |
| | | Atomic | locate bugs (Locate), fix bugs (Fix_Patch), and test cases generation (Test_Patch) | MCQ | 4-64K | 1575 |
| DR | Closed-ended QA | Planning | Select the best next stepwise plan based on trajectory | MCQ | 4-16K | 1031 |
| | | Action | Generate final answer based on trajectory | TC | 4-16K | 350 |
| | Open-ended QA | Plan | Select the best overall report plan from options | MCQ | 4-16K | 298 |
| | | Action | Select the best report based on user's query | MCQ | 32-128K | 214 |
| | | Atomic | Select report statements supported by webpage (Cite) | MCQ | 32-128K | 362 |

## 2.2 CONSTRUCTING APTBENCH-SWE (SOFTWARE ENGINEERING)

We construct APTBench-SWE based on the problem-solving trajectories from two real-world application scenarios: environment setup (EnvSetup) and github issue fixing (IssueFix).

**EnvSetup.** Setting up the environment for code repositories is a highly challenging task (even for human), as it involves understanding the codebase, following instructions provided in the README files, and handling potential errors during the process. We selected research project repositories from nearest ICLR, NeurIPS and ICML conferences, helping to minimize the risk of data leakage. In total, we collected 489 Git repositories as source data.

To evaluate the model's **planning abilities**, we assess *if the base model could select the correct setup plan based on its understanding of repository information.* During the data collection phase,

we regard the repository's README as a single-turn trajectory that a human would follow to conduct environment setup, and use it to generate evaluation questions and answers. In the question generation phase, the question consists of partial information from the repository, including its directory structure and the import statements in each file. We expect the model to choose the correct environment setup plan based on this information. For answer generation, we use an LLM to extract the environment setup steps from the README into a standardized, step-by-step instruction format. We then generate various distractor options by perturbing the correct steps, such as reordering, omitting, or adding redundant steps, to ensure that the steps extracted from the README represent the optimal execution plan. Further details can be found in the Appendix C.

As for the model's **action capability**, we extract all environment setup-related Bash commands from the README using an LLM and *provide the model with the first $T$ steps as input, asking it to generate the $(T+1)$-th command as a text completion task.* The bash command is essentially a tool call to the terminal with correct function name and appropriate parameters.

In the context of environment setup, an important **atomic ability** is handling error cases during setup. During the data collection phase, we filter all issues from the repositories and use an LLM to label and identify those related specifically to environment configuration. We retain issues that are already closed and have highly upvoted solutions, which will be extracted and rewritten into a step-by-step plan using an LLM. Negative plans are constructed in a similar manner to the "planning" part described earlier. Finally, we *ask the model to select the best solution based on the repository's setup document and the issue content that describe the problem.*

**IssueFix.** Solving issues in code repositories is a core capability of code agents. However, mainstream benchmarks such as SWE-bench (Jimenez et al., 2023) require post-trained models, agent frameworks, and Docker environments to perform end-to-end bug fixes, making it difficult to evaluate such capabilities during pretraining. To address this limitation, we design the IssueFix tasks.

We utilize successful trajectories from SWE-Smith (Yang et al., 2025b) as seeds. Each step in this dataset includes a thinking part and an action part. To evaluate the model's **planning ability**, we extract and rewrite the thinking part of the current step to form the ground truth plan, while using plans from other following steps as negative examples because they provide non-logical options.

For assessing **action** execution, we require the model to *generate the Bash command corresponding to the current action step* as a text completion task. The model's input of both planning and action testings consists of the trajectory context up to the current step, including the system prompt, user query, previously executed steps, and environment feedback.

As for atomic capabilities within the IssueFix scenario, we further evaluate the model's **atomic abilities** to locate bugs, fix bugs, and generate test cases targeting the bugs. For this, we use the SWE-Bench-Lite Jimenez et al. (2023) dataset as seed data.

In the *bug localization* task, the input context consists of the problem statement and the files where errors occur (oracle files). We use the code snippet between start and end line number of the gold patch as the ground truth answer, and select other code snippets from the same buggy file as negative examples. The model is prompted to *identify the faulty code snippets* that generate the issue.

For the *bug fix* and *test patch* tasks, we utilize multiple LLMs to generate fix/test patches for SWE-Bench-Lite issues. The patches that could not solve the issue or reproduce the bugs are regarded as the negative choices, respectively. The gold fix patch and test patch in SWE-Bench are used as ground truth answer. The evaluated base models are prompted to *choose from the patch choices*.

Following the aforementioned process, we generate the APTBench-SWE subset of 3,727 questions, as shown in Table 1. More details about statistics are described in Appendix C.3.

## 2.3 CONSTRUCTING APTBENCH-DR (DEEP RESEARCH)

In the deep research scenario, agents search the web to answer user queries, synthesizing information into a final response. These queries fall into two categories: closed-ended questions with clear, concise answers, and open-ended questions that require a comprehensive report. We've created corresponding question sets for both types, with details in Appendix D.

**Closed-ended Question.** We first source queries and corresponding answers from existing benchmark InfoDeepSeek (Xi et al., 2025b). With the framework InfoDeepSeek provides, we generate and filter successful trajectories in a Plan–Action–Feedback format as shown in Appendix D.1.1 with multiple agents. Then, we construct our questions on planning and action abilities as follows:

First, we mainly assess models' **planning abilities** via stepwise planning. There can be multiple correct stepwise plans in this scenario, so we use a multiple-choice format: *given a task and the first $T$ step of trajectory, the model needs to select the most reasonable next-step plan from several choices*. For each successful trajectory, we extract the planning at each step as the correct answer and utilize LLMs to degrade the correct answer and generate incorrect choices. Different degradation rules are employed depending on the type of planning (search, browse, terminate). For example, if the next step is search, incorrect choices might include browsing unrelated documents, ending the task, or repeating a search for an already resolved issue. These options are designed to be unreasonable to ensure the uniqueness of the correct answer. See Appendix D.1.2 for more details.

Then, evaluating the **action ability** primarily involves generating the correct answer based on the user's query and full search and browsing trajectory. We do not consider tool usage at each step, as this is already covered in the planning tasks mentioned above. Given that the answers to closed-ended questions are relatively clear and concise, we use a text completion format. To facilitate the evaluation of base models, we employ LLMs to shorten and summarize the correct answer, standardizing the format (*e.g.*, time and numbers) and removing excessively lengthy answers.

**Open-ended Questions.** We collect open-ended questions from existing benchmarks, *e.g.*, Deep-Research Bench (Du et al., 2025a) and Researchy Questions (Rosset et al., 2024). We gather reports from high-performing Deep Research Agents on these open-ended questions to prepare seed data.

First, regarding **planning abilities**, we mainly focus on overall planning. As the information-seeking process for open-ended questions involves various aspects, it entails the model's overall planning ability to break down and organize tasks at a high level. Such planning does not have a single optimal solution, so we adapt a multiple-choice format: *asking the model to select the best overall plan from the choice*. We utilize the standard plan from the Researchy Questions as the correct answer and create incorrect choices by degrading the standard plan through strategies like swapping the order of sequential sub-plans, randomly deleting some sub-plans, or adding irrelevant sub-plans.

Next, in terms of **action abilities**, we primarily focus on the model's ability to generate final reports, as it is challenging to assess the correctness of intermediate tool usage. Since base models struggle with instruction-following to generate long reports, we also adapt a multiple-choice format: *the model must select the best report from the options for a given query*. We use the collected reports from high-performing Deep Research Agents as the correct answers. Then, we leverage LLM to degrade them by disrupting key aspects such as accuracy, logic, readability, and alignment with user requirements, generating lower-quality reports as incorrect choices. Since each option is a report, these questions are typically very lengthy.

Finally, regarding **atomic abilities**, we focus on the model's ability to cite. Although open-ended questions lack standard answers, it is still crucial for the report to be appropriately supported by factual evidence. Thus, the model's ability to correctly cite relevant sources to support its statements is essential. We still follow a multiple-choice format: *given a report, a cited webpage content, and options that are statements from the report, the model must identify which statements in the report are supported by the web page*. We leverage LLMs to extract all statements that cite the given webpage as correct answers, while LLMs also generate incorrect choices by extracting statements unrelated to the webpage. Note that a report may cite the same webpage multiple times, so this type of question may have multiple correct answers.

After above process and human validation, we obtain a total of 2,255 questions, containing both English and Chinese ones, as shown in Table 1. See Appendix D.3 for more details about statistics.

## 2.4 EVALUATION SETTINGS

Since base models often struggle to follow instructions and format constraints, we employ **few-shot prompting** to ensure accurate extraction of model outputs. Most tasks use 3-shot prompting as examples, detailed evaluation prompts and examples can be found in Appendix E.

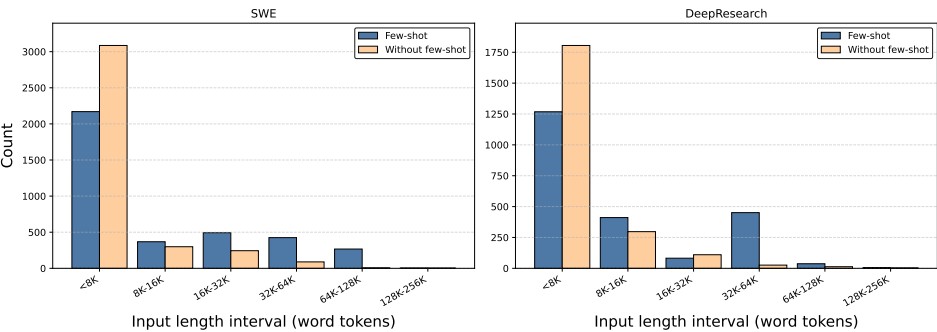

Figure 3: The prompt length distribution of APTBench.

For multi-choice questions (MCQ), we use Accuracy (ACC) as the evaluation metric. For text completion (TC) tasks, Exact Match (EM) and ROUGE scores are utilized. In text completion tasks under the SWE scenario, we primarily focus on EM, as this is essentially tool calling, requiring the accurate tool invoking and precise arguments, thus demanding exact match. In contrast, for DR scenarios that involve answer summarization, where variations in expression are acceptable, we consider both EM and ROUGE scores.

### 2.5 FEATURES OF APTBENCH

APTBench is especially designed for base model evaluation during pre-training stage, and provides a simple and economic way of probing the base models' potential on agentic tasks.

**Economic and Accurate Way to Evaluate Base Models.** APTBench makes it possible to economically evaluate a model's agentic potential during the pre-training phase. At this stage, we typically use general pre-training metrics as a surrogate for agent capabilities. However, as analyzed earlier in Figure 1 and Appendix B, such general metrics often fail to accurately reflect a model's true agentic abilities after post-training. Moreover, the nature of the pre-training phase renders most existing end-to-end agent benchmarks inapplicable.

**Generated from Long & Multi-turn Trajectories.** APTBench is primarily built from real problem-solving trajectories by either agents or humans, which are characterized by long sequences and multiple interaction rounds. This long agent context allows us to jointly evaluate both the model's agentic capabilities and its long-context processing abilities. The length distribution of APT-Bench is shown in Figure 3, including input length w/ and w/o few-shot prompting. As shown in the figure, a significant portion of APTBench contexts exceed 16K tokens. When few-shot prompting are included, the prompt length increases substantially.

**Scalability.** Our proposed methodology for constructing APTBench is comprehensive and systematic, so it can be easily extended to other agent domains. Moreover, the dataset can be continuously updated using the same approach with refreshed seed data, such as new GitHub repositories, agent models, and frameworks, helping to mitigate the risk of data leakage. In addition, our method can also be applied to large-scale synthesis of agentic pretraining data.

## 3 EXPERIMENTS

We evaluate several representative open-source models, including the Qwen3 series (Yang et al., 2025a) (1.7B/4B/8B/30BA3B) [2], Llama3.2-3B, Llama4-Scout (MetaAI, 2025), DeepSeek (V3/V3.1) (Liu et al., 2024; DeepSeek-AI, 2025), SmolLM3-3B (Bakouch et al., 2025), Seed-OSS-36B (ByteDance-Seed, 2025b), Gemma3-27B (Gemma-Team et al., 2025), as well as GLM-4.5, GLM-4.5-Air (Zeng et al., 2025) and Kimi-K2 (Kimi-Team, 2025). See Appendix E for more details. The tested base models cover the range from small dense models to very large MoEs, with some of them undergone agent-oriented pre-training (Zeng et al., 2025). The performance of small

---

[2]We didn't include 32B dense and 235B MoE base models as they are not open-sourced.

Table 2: Base model performance on APTBench-SWE. All models listed are base models. The Action (LLM) columns are the LLM-as-a-judge results of comparing the model output with ground-truth bash commands.

| Model Names | EnvSetup | | | | IssueFix | | | | | | AVG |
|---|---|---|---|---|---|---|---|---|---|---|---|
| | Plan (ACC) | Action (EM) | Action (LLM) | Handle_Error (ACC) | Locate (ACC) | Fix_Patch (ACC) | Plan (ACC) | Action (EM) | Action (LLM) | Test_Patch (ACC) | |
| Qwen3-1.7B | 35.47 | 19.93 | 20.39 | 17.01 | 26.15 | 25.43 | 27.16 | 17.84 | 23.24 | 25.19 | 24.27 |
| Qwen3-4B | 71.85 | 31.64 | 33.58 | 61.90 | 28.98 | 25.86 | 41.98 | 22.82 | 25.73 | 25.00 | 38.75 |
| Qwen3-8B | 76.43 | 35.61 | 38.19 | 61.22 | 25.09 | 25.43 | 52.67 | 27.80 | 31.54 | 28.68 | 41.62 |
| Llama3.2-3B | 14.19 | 9.50 | 10.33 | 25.17 | 23.32 | 29.31 | 30.45 | 25.31 | 28.22 | 24.62 | 22.73 |
| SmolLM-3B | 27.92 | 20.48 | 23.62 | 17.01 | 23.67 | 29.31 | 32.51 | 22.41 | 25.31 | 21.42 | 24.34 |
| Qwen3-30BA3B | 74.83 | 33.03 | 34.96 | 59.86 | 22.26 | 32.33 | 53.50 | 29.46 | 32.78 | 27.55 | 41.60 |
| Seed-OSS-36B | 89.24 | 39.76 | 42.80 | 83.67 | **33.57** | 34.91 | **60.91** | 28.22 | 32.37 | **29.15** | 49.93 |
| Gemma3-27B | 52.40 | 37.18 | 39.76 | 49.66 | 28.98 | 26.72 | 46.50 | 11.20 | 14.11 | 24.91 | 34.69 |
| Llama4-Scout-109BA17B | 56.98 | 36.07 | 38.38 | 31.97 | 25.09 | 24.14 | 50.62 | 28.63 | 32.37 | 25.38 | 34.86 |
| GLM4.5-Air-106BA12B | 70.71 | 39.48 | 42.62 | 70.07 | 28.98 | 32.33 | 39.92 | 26.97 | 31.54 | 26.32 | 41.85 |
| GLM4.5-355BA32B | 78.49 | **43.17** | **46.49** | 74.15 | 26.50 | 37.07 | 33.74 | 0.00 | 0.41 | 29.06 | 40.27 |
| DeepSeek-V3-671BA37B | 85.58 | 42.25 | 45.30 | 78.91 | 28.33 | 50.00 | 55.14 | 28.63 | 30.71 | 24.35 | 49.15 |
| DeepSeek-V3.1-671BA37B | **90.16** | 42.07 | 45.11 | 82.31 | 30.04 | 43.97 | 58.02 | 28.63 | 30.71 | 24.91 | 50.01 |
| Kimi K2-1TA32B | 83.98 | 40.68 | 44.00 | **85.71** | **33.57** | **65.95** | 52.26 | **30.29** | **33.61** | 28.02 | **52.56** |

Table 3: Base model performance on APTBench-DR. All models listed are pre-trained base models. The Act_EN/Act_ZH (LLM) columns are the LLM-as-a-judge results of comparing the model output with ground-truth answers.

| Models | Closed-ended Question | | | | | | | | Open-ended Question | | | | | Avg |
|---|---|---|---|---|---|---|---|---|---|---|---|---|---|---|
| | Plan_EN (ACC) | Plan_ZH (ACC) | Act_EN (EM) | Act_EN (ROUGE-1) | Act_EN (LLM) | Act_ZH (EM) | Act_ZH (ROUGE-1) | Act_ZH (LLM) | Plan_EN (ACC) | Cite_EN (ACC) | Cite_ZH (ACC) | Act_EN (ACC) | Act_ZH (ACC) | |
| Qwen3-1.7B | 53.09 | 36.45 | 30.66 | 46.03 | 52.83 | 26.09 | 42.26 | 51.45 | 20.13 | 9.83 | 0.53 | 27.27 | 21.36 | 28.52 |
| Qwen3-4B | 72.48 | 68.59 | 30.19 | 49.69 | 55.19 | 31.88 | 51.11 | 60.87 | 56.04 | 20.23 | 3.70 | 37.27 | 24.27 | 40.50 |
| Qwen3-8B | 73.94 | 72.90 | 31.60 | 48.41 | 54.72 | 36.96 | 54.66 | 63.77 | 55.03 | 15.61 | 8.47 | 39.09 | 29.13 | 42.35 |
| Llama3.2-3B | 41.69 | 30.94 | 24.53 | 39.45 | 42.45 | 16.67 | 35.27 | 34.78 | 13.76 | 14.45 | 2.65 | 27.27 | 29.13 | 25.07 |
| SmolLM-3B | 48.70 | 25.66 | 25.00 | 41.69 | 44.81 | 31.16 | 45.29 | 45.65 | 16.11 | 3.47 | 0.53 | 33.64 | 28.16 | 27.22 |
| Qwen3-30BA3B | 73.13 | 68.82 | 32.55 | 50.47 | 57.08 | 36.23 | 51.34 | 61.59 | 63.09 | 16.76 | 8.47 | 54.55 | 45.63 | 45.55 |
| Seed-OSS-36B | 81.60 | 77.94 | 43.87 | 61.95 | 65.57 | 42.03 | 63.09 | 69.57 | 86.24 | 42.20 | 10.58 | 90.00 | 77.67 | 61.56 |
| Gemma3-27B | 73.13 | 59.95 | 40.57 | 57.45 | 63.21 | 37.68 | 55.62 | 63.04 | 45.30 | 18.50 | 5.29 | 28.18 | 32.04 | 41.25 |
| Llama4-Scout-109BA17B | 78.01 | 67.63 | 36.32 | 55.11 | 58.96 | 31.88 | 54.51 | 56.52 | 50.34 | 30.64 | 10.05 | 34.55 | 26.21 | 43.20 |
| GLM4.5-Air-106BA12B | 74.27 | 67.87 | 41.04 | 58.58 | 61.79 | 44.93 | 61.06 | 61.59 | 67.11 | 35.84 | 7.41 | 48.18 | 58.25 | 51.32 |
| GLM4.5-355BA32B | 85.50 | 79.14 | 44.81 | 65.00 | 73.11 | **48.55** | 67.12 | 71.74 | 84.90 | 31.21 | 5.82 | 57.27 | 75.73 | 58.64 |
| DeepSeek-V3-671BA37B | 82.31 | 75.54 | 44.34 | 65.03 | 69.81 | 42.03 | 64.70 | 68.84 | **88.26** | 45.09 | 21.16 | 84.55 | 63.11 | 61.47 |
| DeepSeek-V3.1-671BA37B | 84.04 | 76.74 | **45.28** | **65.51** | 70.28 | 42.75 | 64.47 | 68.12 | **88.26** | **54.34** | **25.40** | **94.55** | **89.32** | **66.42** |
| Kimi K2-1TA32B | **86.64** | **79.38** | 43.87 | 63.98 | 67.45 | 43.48 | **69.15** | **71.01** | 79.19 | 15.03 | 12.17 | 40.91 | 50.49 | 53.12 |

models is also important, as they could be suitable for agent scenarios (Belcak et al., 2025; Shang et al., 2025). The SWE and DR results are shown in Table 2 and Table 3, respectively.

## 3.1 OBSERVATIONS

**Emergence happens at a critical model size.** By comparing the performance of the Qwen3 series models including 1.7B, 4B, 8B and 30BA3B MoE, we observe a clear performance gap between the 1.7B model and the larger three ones. Specifically, the average scores on APTBench-SWE are 24.27, 38.75, 41.62, and 41.60, respectively; and on APTBench-DR, the scores are 28.52, 40.50, 42.35, and 45.55. In contrast, the latter three models exhibit relatively similar performance. These results indicate that the emergence of agent capabilities requires the model to exceed a fundamental parameter size threshold. Models that are too small fail to acquire such capabilities, making them unsuitable as base models for agent systems.

**Medium-sized model could achieve outstanding scores.** In Table 2 and Table 3, we could observe very competitive results from Seed-OSS-36B. It achieves nearly or even better performance compared to DeepSeek-V3.1 and Kimi-K2 on many tasks of APTBench. These two large MoEs have same level of activated parameters as Seed-OSS-36B, indicating 30B level of parameters could be a sweet spot for agent models.

**Training data is the most essential part of agentic pre-training.** By examining 3–4B dense models (i.e., Qwen3-4B, LLaMA3.2-3B, SmolLM3-3B) and 100B MoE models (i.e., GLM4.5-Air-106BA12B, Llama4-Scout-109BA17B), we observe significant differences in agent evaluation performance. For instance, Qwen3-4B outperforms SmolLM3-3B by 59.2% and 48.8%, on SWE and DR respectively. And GLM4.5-Air leads Llama4-Scout by 20.1% and 18.8%. These models share similar architectures and parameter scales, suggesting that the primary factor driving this performance gap lies in whether their pre-training data has been optimized for agent-centric scenarios Zeng et al. (2025). This highlights the critical role of data quality and task alignment in developing effective agent base models, even beyond model size or architecture.

### 3.2 DISCUSSIONS

**APTBench is closely correlated with final agentic capabilities.** Similar to the approach in Figure 1, we present the performance of several models' base versions on APTBench alongside the performance of their instruct versions on SWE-bench Verified in Figure 4. Since there is no widely adopted agent framework or benchmark for deep research, we use SWE-bench Verified as an accepted proxy for estimating agent performance. Compared with the general benchmarks (MMLU, EvalPlus, and GSM8K) in Figure 1, the results of APTBench-SWE and APTBench-DR show a much stronger, positive correlation with SWE-bench. This suggests that APTBench reflect the agentic potential of base models and offering more useful guidance for base model training. Note that some models shown in Figure 1 do not appear in Figure 4 because their corresponding base models are not publicly available, *e.g.*, LongCat-Flash, Seed-1.6-Thinking, Qwen3-235B, and Qwen3-32B.

**The long-context capability of models also affects their performance on APTBench.** We observe that after removing tasks with very long context, *e.g.*, the plan and action sub-tasks in IssueFix, as well as the citation and action sub-tasks in Open-ended Question, APTBench exhibits a stronger correlation with the downstream agent evaluation results, as shown in Figure 4(b) and (d). This could be attributed to some models' less robust capacity to handle long-context, which limited their performance on these tasks. This finding highlights the importance of further enhancing pre-training on data with long and complex trajectories, as they are critical for robust agent performance.

### 4 RELATED WORKS

**Benchmarks for Base Model.** Existing benchmarks for evaluating base models can be divided into three categories: general knowledge, math, and code. The general benchmarks mainly assesses language understanding and knowledge mastery of base models, including MMLU (Hendrycks et al., 2020), MMLU-Pro (Wang et al., 2024b), BBH (Suzgun et al., 2022), SimpleQA (Wei et al., 2024), GPQA (Rein et al., 2024), and SuperGPQA (Du et al., 2025c). The math benchmarks focus on evaluating the model's mathematical reasoning ability, including GSM8K (Cobbe et al., 2021), MATH (Hendrycks et al., 2021), CMATH (Wei et al., 2023), and others. The code benchmarks includes HumanEval (Chen et al., 2021), MBPP (Austin et al., 2021), EvalPlus (Liu et al., 2023a) (average of HumanEval+, and MBPP+), LiveCodeBench (Jain et al., 2024), CRUXEval (Gu et al., 2024), and more. Due to the base model's relatively poor instruction-following ability, most of the problems in these benchmarks are multiple-choice or completion task, and the majority use few-shot prompts to guide the output format. However, these benchmarks have weak relevance to real-world agent tasks and are difficult to use for evaluating a model's potential in agentic tasks.

**Benchmarks for Agent.** To evaluate the capabilities of LLM Agents, researchers have developed various specialized benchmarks. Some of these benchmarks target core abilities of agents, such as planning and multi-step reasoning (Valmeekam et al., 2023; Kokel et al., 2025), tool usage (Qin et al., 2023; Patil et al., 2024), and memory (Packer et al., 2023; Zhong et al., 2021). Others are designed to simulate real-world tasks and scenarios, including deep research (Wei et al., 2025;

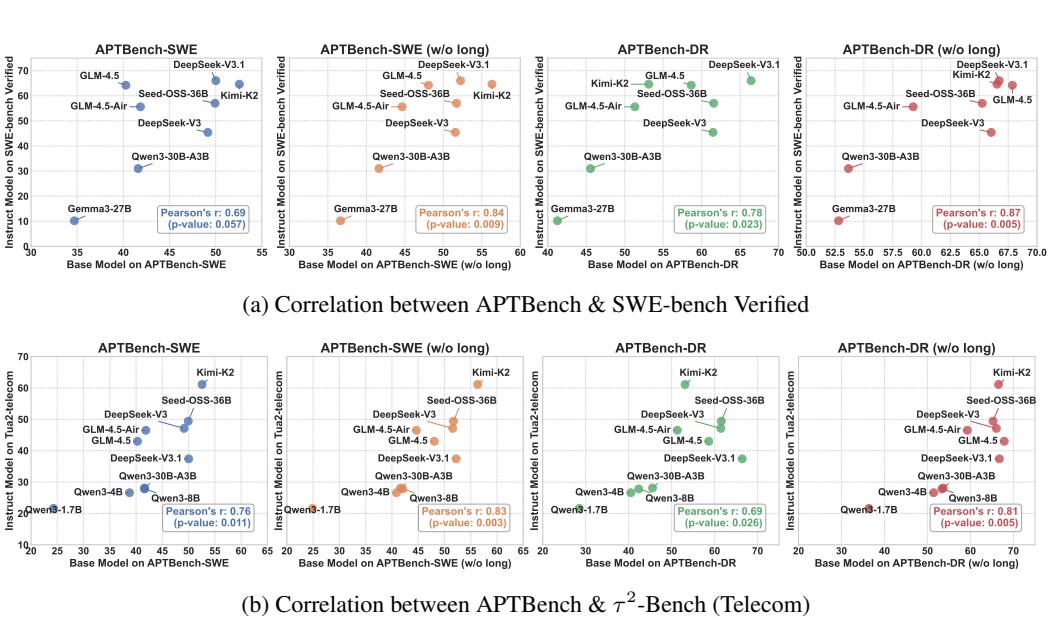

(a) Correlation between APTBench & SWE-bench Verified

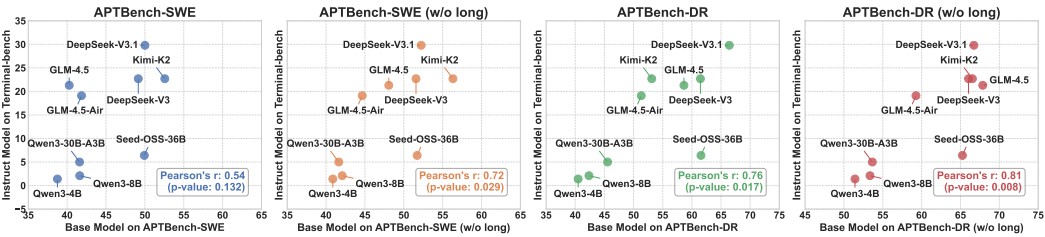

(b) Correlation between APTBench & $\tau^2$-Bench (Telecom)

(c) Correlation between APTBench & Terminal Bench

Figure 4: The correlation between model's performance on agent benchmarks (SWE-bench Verified) and our APTBench (SWE, SWE w/o long-context tasks, DR, and DR w/o long-context tasks). The high Pearson correlation coefficient (r) and low p-values indicate a strong correlation.

Du et al., 2025a; Xi et al., 2025b), software engineering (TTB-Team, 2025; Jimenez et al., 2023; Yang et al., 2024), web automation (Barres et al., 2025; Zhou et al., 2023; Yao et al., 2022; Deng et al., 2023), operating systems (Xie et al., 2024; Rawles et al., 2024), and scientific research (Chen et al., 2024). There are also some benchmarks that integrate tasks from multiple scenarios (Barres et al., 2025; Galileo, 2025; Liu et al., 2023b; Mialon et al., 2023). Currently, these benchmarks are focused on post-trained models, which are capable of following complex instructions, utilizing external tools, and completing tasks through multi-turn interactions. However, base models lack instruction-following abilities and cannot complete tasks end-to-end, so it is challenging to assess them on these benchmarks.

## 5 CONCLUSION

In this paper, we propose APTBench, the first benchmark specifically designed to evaluate the agent potential of pre-training base language models. Compared to general pre-training evaluation suites, APTBench demonstrates stronger correlation with downstream agent tasks. This benchmark provides a dedicated evaluation tool for agent-oriented pretraining, facilitating deeper research into agent-relevant capabilities during the pre-training phase and helping advance the development of more capable agent models.

## REPRODUCIBILITY STATEMENT

To facilitate the full reproducibility of our work, we provide comprehensive details on our methodology and experimental setup. A complete description of our data construction pipeline, including construction approach and their construction prompts for each tasks, is included in Section 2.1, Appendix C and D. Furthermore, Section 2.4 and Appendix E contain all relevant experimental details, such as model specifications, hyperparameters, and evaluation prompts. For review purposes, our benchmark data is uploaded as the supplementary material; Code & data will be made publicly available in the future.

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

TABLE OF CONTENTS

## A    THE USE OF LLMS

The use of large language models (LLMs) in this paper was strictly limited to language refinement. The models were employed to improve clarity, correct grammar, and assist with translations to enhance the readability and accessibility of the manuscript. They were not used for research ideation, data analysis, or the generation of any core content.

## B    ANALYSIS OF EXISTING GENERAL AND AGENT BENCHMARKS

In this section, we analyze the relationship between the performance of existing general benchmarks for evaluating base models and agent benchmarks for evaluating instruct models. We selected three representative general benchmarks:

- **MMLU** (Wang et al., 2024b): which evaluates LLMs' language understanding and knowledge across a broad range of challenging tasks.
- **GSM8K** (Cobbe et al., 2021): a dataset containing 8.5K high-quality, linguistically diverse grade school math word problems, designed for question answering on basic mathematical problems requiring multi-step reasoning.
- **EvalPlus** (Chen et al., 2021): a rigorous evaluation framework for assessing LLM performance in code generation tasks, averaging the results of HumanEval+ and MBPP+.

Additionally, we chose three widely recognized agent benchmarks:

- **SWE-bench Verified** (Jimenez et al., 2023): a human-validated subset of SWE-bench that reliably evaluates AI models' ability to solve real-world software issues.
- **Terminal-Bench** (TTB-Team, 2025): a collection of tasks and an evaluation harness that helps agent developers quantify their agents' mastery of terminal commands.
- **Tua2-Bench** (Barres et al., 2025): a simulation framework for evaluating customer service agents across various domains such as retail and airline.

We selected several representative models, including DeepSeek-V3 (Liu et al., 2024), DeepSeek-V3.1 (DeepSeek-AI, 2025), Qwen3-235B-A22B (Yang et al., 2025a), Qwen3-30B-A3B (Yang et al., 2025a), Qwen3-32B (Yang et al., 2025a), Llama-4-Maverick (MetaAI, 2025), Kimi-K2-Instruct (Kimi-Team, 2025), seed-oss-36B (ByteDance-Seed, 2025b), Seed1.6-Thinking-0715 (ByteDance-Seed, 2025a), GLM-4.5 (Zeng et al., 2025), LongCat-Flash (Meituan et al., 2025), and Devstral-Small-1.0 (Mistral-AI, 2025). We evaluate the above models' base versions on general benchmarks and instruct versions on agent benchmarks, as show in Figure 1, 5, 6 and 7. The performance of the models comes from their publicly available technical reports or model cards, so some data may be missing, which results in a varying number of points in each figure. To minimize the impact of different agent frameworks, we ensured that the instruct model results on each agent benchmark came from the same framework whenever possible. For example, most results on SWE-bench Verified are based on OpenHands, results on Terminal-Bench are based on Terminus, and results on Tua2-bench are based on the official agent framework. From the figures above, we can make the following observations:

First, the performance of base models on general benchmarks (MMLU, EvalPlus, GSM8K) may have limited correlation with their instruct model performance on agent benchmarks (SWE-bench Verified, Terminal-bench, Tua2-retail, Tua2-airline), as seen in the Figure 1, 5, 6 and 7. For example, DeepSeek-V3 performs poorly on EvalPlus but performs very well on Tua2-retail and Tua2-airline. Additionally, in the GSM8K subplot, GLM-4.5 has relatively low scores in mathematical reasoning, below 80. However, its performance on SWE-bench Verified is very high, ranking similarly to DeepSeek-V3.1 and Kimi-K2, which scored above 92 on GSM8K. Qwen3-235B-A22B has the highest score on MMLU but performs worse on Terminal-Bench than many models with lower MMLU scores. This could be because general benchmarks (such as MMLU, GSM8K, and EvalPlus) primarily assess static, single-turn knowledge, logic, and coding abilities. In contrast, real-world agent tasks (like SWE-bench Verified and Terminal-bench) require models to engage in multi-turn interactions, planning, and flexible responses to dynamic feedback from external environ-

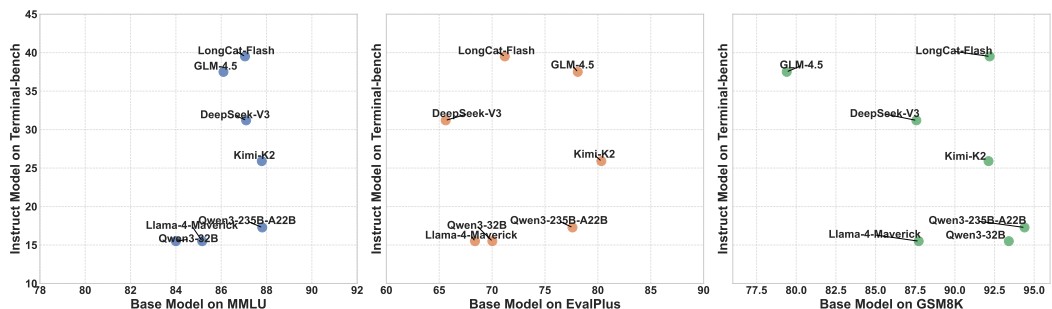

Figure 5: The correlation between model's performance on general benchmarks (MMLU, EvalPlus, and GSM8K) and agent benchmarks (Terminal-Bench).

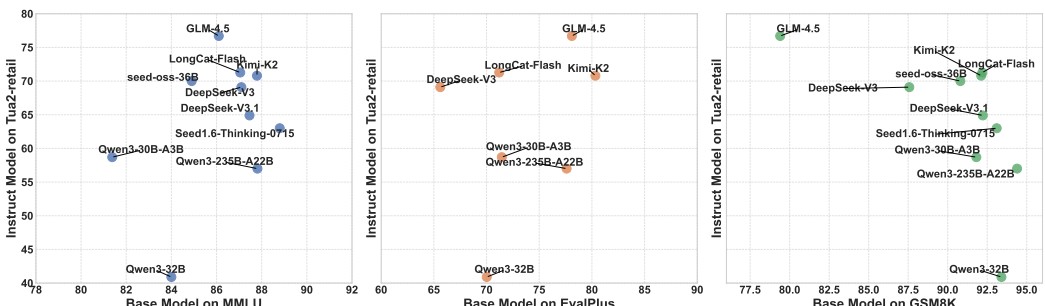

Figure 6: The correlation between model's performance on general benchmarks (MMLU, EvalPlus, and GSM8K) and agent benchmarks (Tua2-Retail).

ments. These dynamic, multi-step decision-making and planning abilities are difficult for general benchmarks to effectively evaluate.

Secondly, the score differences on general benchmarks are relatively small, whereas agent benchmarks are more effective in distinguishing model performance. For example, in the MMLU and GSM8K figures, most models' scores are clustered in a narrow range of 80 to 90. In contrast, on SWE-bench Verified, the model scores range more widely, from around 20 to over 60. Similarly, on Terminal-bench, model scores range from 15 to over 40, with significant differences. This could be because many mainstream large language models have already reached a high level of general capabilities (such as knowledge, language understanding, and mathematics), causing their scores to converge. However, in complex and dynamic agent tasks, even small differences in model architecture, training data, and reasoning abilities are amplified, leading to larger performance gaps. While many models have converged in general capabilities, the real differences between them emerge when

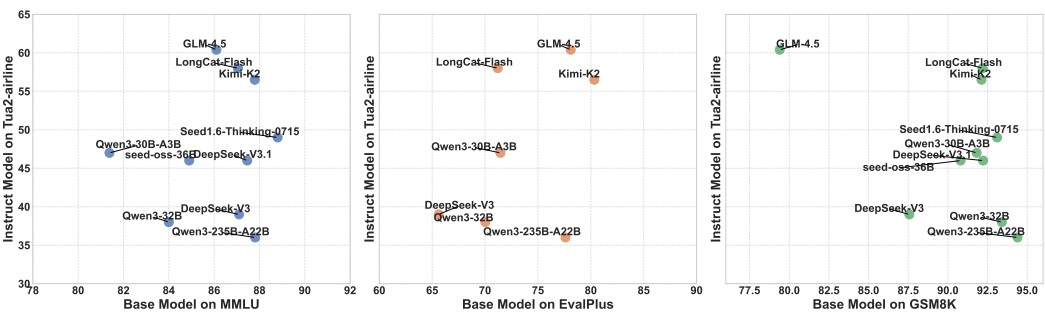

Figure 7: The correlation between model's performance on general benchmarks (MMLU, EvalPlus, and GSM8K) and agent benchmarks (Tua2-Airline).

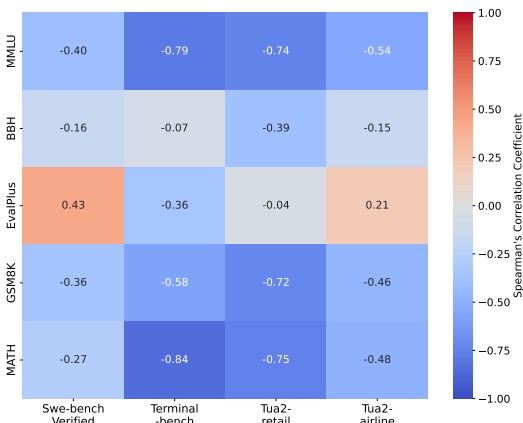

Figure 8: The Pearson correlation coefficient between model's performance on general benchmarks (MMLU, BBH, EvalPlus, MATH, and GSM8K) and agent benchmarks (SWE-bench Verified, Terminal-Bench, as well as Tua2-retail and Tua2-airline).

faced with complex agent tasks that require advanced reasoning and planning abilities. Compared to general benchmarks, agent benchmarks are more effective in differentiating model performance.

Lastly, although general benchmark scores are not good predictors of agent performance, a few models consistently perform well across all types of agent benchmarks. This suggests that these models may possess a universal and strong "agentic ability." On the SWE-bench Verified benchmark, Kimi-K2 and GLM-4.5 are in the top tier, exhibiting excellent performance. On the Tua2-retail and Tua2-airline benchmarks, GLM-4.5, LongCat-Flash, and Kimi-K2 also maintain high rankings, far outpacing other models. On the Terminal-bench benchmark, LongCat-Flash and GLM-4.5 again score the highest. This consistency across tasks indicates that these models not only excel in specific coding or business process tasks but also possess deeper planning, reasoning, and dynamic adaptation capabilities, allowing them to flexibly handle different types of complex tasks.

Additionally, we computed the Pearson correlation coefficient between model performance on general benchmarks and agent benchmarks using the data above, and the results are presented in Figure 8. For the general benchmarks, we selected MMLU and BBH for knowledge, EvalPlus for code generation, and GSM8K and MATH for mathematics. For the agent benchmarks, we used SWE-bench Verified, Terminal-Bench, as well as Tua2-retail and Tua2-airline. As shown in Figure 5, the correlation between these benchmarks is quite weak, with most of the correlations being negative. This suggests that it is difficult to gauge a model's potential in agent tasks based solely on its performance on general benchmarks. The negative correlations further emphasize the challenge of relying on general benchmark performance as an indicator of success in agent-specific tasks, highlighting the distinct nature of the skills required for each.

## C    CONSTRUCTION DETAILS OF APTBENCH-SWE (SOFTWARE ENGINEERING)

### C.1    ENVSETUP

#### C.1.1    TASK AND TRAJECTORY COLLECTION

For the EnvSetup scenario, we collected GitHub repositories corresponding to papers from ICML 2025, ICLR 2025, and NeurIPS 2024. After filtering for repositories that contain a complete README file in the root directory, we obtained a total of 489 repositories.

The README files from these repositories serve as the primary source of seed data for evaluating planning and action capabilities, while the GitHub issues are used as the main seeds for evaluating the error handling atomic skill. In the following sections, we provide detailed descriptions of each component.

### C.1.2 PLANNING ABILITIES

For each repository, we use an LLM to extract a step-by-step execution plan and the corresponding bash commands from the README file. This process yields the ground truth plan and its associated execution commands for setting up the repository environment. The prompt used for this step is shown in Prompt 1.

Subsequently, we apply Prompt 2 to deliberately perturb the ground truth plan, introducing modifications including removing steps, inserting redundant steps, or shuffling the order of steps. These corrupted plans are used as negative examples in our multiple-choice question (MCQ) format.

---

**Prompt 1: Step-by-Step Plan & Action Generation**

You are an expert assistant specializing in software documentation. Your task is to analyze the provided README file and extract a step-by-step guide on how to run the repository. From the content of the README file below, generate a set of instructions.
README Content: {README_FILE}
Instructions:

1. Read through the entire README file to understand the setup and execution process.

2. Identify all the necessary steps to run the project.

3. Give a list of plans, providing a clear description of what each step accomplishes.

4. Give a list of commands that need to be executed, each command corresponds to one plan, if the plan does not have a executing command, return None.

5. Format the output as two numbered list of steps (First is plans list, second is commands list), starting with "step1:"

6. Directly return the step-by-step plan and execution commands with nothing else.

Example Output Format:
Execution plans:
step1: [Description of the first step];
step2: [Description of the second step];
stepN: [Description of the nth step];
//////
Execution commands:
step1: [cmd1,cmd2];
step2: [None];
stepN: [cmd1];

---

**Prompt 2: Generation of Error Plans**

Your task is to act as a "Chaos Engineer" for procedural instructions. I will provide you with a correct, step-by-step "ground truth" execution guideline. Your goal is to generate a specified number of distinct, erroneous execution flows based on this ground truth. These error flows should be plausible yet incorrect, simulating various ways a user might misunderstand or incorrectly follow the instructions.
Ground Truth Plan Guideline: {Ground_Truth}
Instructions for Generating Error Flows: Generate 2-3 error flows for each following error types:

1. Dependency Violation Shuffle: Reorder the steps so that a step is executed before a step it depends on.

2. Critical Step Omission: Remove one or more essential steps from the procedure without which the final outcome cannot be achieved correctly.

3. Harmful Step Addition: Insert a new step that is counterproductive, dangerous, or directly conflicts with other steps.

Renumber all the remaining step in the correct order.

---

Output Format: For each generated error flow, use the following format:
Error Flow X:
(Title of the Error Type);
(What you did to the ground truth);
(The generate error flow)
—
Error Flow Y:
(Title of the Error Type);
(What you did to the ground truth);
(The generate error flow)

Directly return the error plans with nothing else.

### C.1.3 ACTION ABILITIES

For the action abilities, we use the ground truth bash command before the $T$-th step and let the model to write what should be the next command as a text completion task. The ground truth command steps are also extracted from the repo's README using prompt 1. Due to the output instability of the base model, we filter out commands that exceed 20 tokens in length.

### C.1.4 ATOMIC ABILITIES

For the atomic ability of EnvSetup, we ask the tested model to select the correct plan when the setup meets error. For the error here, we use the issues from the repository. We filtered for closed GitHub issues that have more than three responses and contain at least one answer with a positive number of likes. It gives us 640 issues and we further keep the issues that are about environment setup. The full content of each issue thread is then passed to an LLM, which determines whether the issue has been successfully resolved, and, if so, generates a summary execution plan based on the content of the most upvoted response. After the pre-process, we get 147 issues.

Using the same approach as shown in Prompt 2, we generate incorrect versions of the plan by introducing modifications such as step removal, redundant additions, or step reordering. These incorrect plans serve as negative choices in a multiple-choice question (MCQ) format, just like in the planning task.

## C.2 ISSUEFIX

### C.2.1 TASK AND TRAJECTORY COLLECTION

For the IssueFix scenario, we mainly use SWE-Bench Jimenez et al. (2023) as the seed dataset. We directly use the trajectory from SWE-Smith-Trajectory SWE-Bench (2025) that successfully resolve the issues from SWE-Bench.

### C.2.2 PLANNING ABILITIES

For the planning abilities, we use the trajectory before step $T$ as the context and retrieve the LLMs thought at $T + 1$ as the ground-truth next step plan. The negative plans are sampled from the following steps after $T + 1$. The example of the ground-truth plan and negative ones are shown in Example 1.

---

**Example 1: Example for Next-Step Plan of IssueFix**

GT_PLAN: I need to run our reproduction script again to see if the issue is fixed.
NEGATIVE_PLAN1: I need to fix the 'estimate_type' method to return the correct integer values.
NEGATIVE_PLAN2: I need to also check if there are any other tests that might be affected by our changes.
...

---

Table 4: The statistics for APTBench-SWE (Software Engineering)

| | EnvSetup | | | IssueFix | | | | |
|---|---|---|---|---|---|---|---|---|
| | Planning | Action | ErrorHandle | Planning | Action | BugLocate | FixPatch | TestPatch |
| Size | 437 | 1084 | 147 | 243 | 241 | 283 | 232 | 1060 |

### C.2.3 ACTION ABILITIES

The action ablity is also formulated as a text completion (TC) task, given the previous trajectory and the toolset, the model need to write the next command. The tool set here includes bash terminal, str_replace_editor and submit (represents the issue has been fixed and the model submit the answer). The model's next action is one of these three tool calls.

### C.2.4 ATOMIC ABILITIES

As for the atomic abilities for IssueFix scenario, we incorporate BugLocate, FixPatch and TestPatch tasks, meaning locating the bug snippet, selecting the fixing patch and test patch from the choices. For these tasks, we use SWE-Bench-Lite Jimenez et al. (2023) as our source data.

**BugLocate** We firstly extract the start and end range of the code modification from the git diff information provided in the gold fixing patch. This range is considered the buggy code segment, and it serves as the positive example in our MCQ evaluation. To increase the difficulty of the task, we then sample three additional code segments that overlap with the original range as challenging negative examples. The model input includes both the issue context and the relevant files from the repository, providing the necessary information to identify the correct buggy code segment.

**FixPatch** For FixPatch task, the model is asked to select the correct fixing patch for the given issue. We use the oracle setup of the SWE-Bench Jimenez et al. (2023) which provide the oracle bug file to the LLMs and ask them to generate a fix patch. After getting multiple fix patches for each issue, we evaluate all the fix patches and preserve the failed ones, which are then utilized as the negative choices.

**TestPatch** Similarly to the FixPatch task, we use different LLMs to generate different test patches that could reproduce the issue and select the ones that failed to do so as the negative choices.

### C.3 STATISTICS

We compiled statistics on the number of different types of questions in the software engineering scenario, as shown in Table 4. All questions in this scenario are in English.

## D CONSTRUCTION DETAILS OF APTBENCH-DR (DEEP RESEARCH)

In the deep research scenario, the agent is required to actively search for and browse information on the web in order to address complex user queries. Beyond simply retrieving isolated facts, the agent must aggregate information from multiple sources, evaluate the credibility of these sources, and synthesize the relevant content into a coherent final answer. The user queries in this scenario can be broadly divided into two categories. The first category is closed-ended questions, which admit clear and concise answers, often grounded in factual knowledge or a specific piece of evidence. These queries typically test the agent's ability to locate precise information and respond with accuracy and brevity. The second category is open-ended questions, which do not have a single definitive answer. Instead, they require the agent to conduct more extensive research, integrate evidence from diverse perspectives, and produce longer, report-style responses that demonstrate reasoning, organization, and comprehensiveness.

## D.1 Closed-ended Question

### D.1.1 Task and Trajectory Collection

For closed-ended questions, we primarily draw on challenging queries and human-verified answers from the existing benchmark InfoDeepSeek (Xi et al., 2025b). These queries are designed to require a search agent to engage in multiple rounds of search and web-browsing for information gathering. Based on the agent framework provided by InfoDeepSeek, we generate browsing trajectories in a Plan–Action–Feedback format. Here, Actions mainly consist of tool invocations, including search tools (Google & DuckDuckGo APIs), a web-browsing tool, and a termination tool. Feedback refers to the search results returned by the APIs and the webpage content retrieved by the browsing tool. Finally, the agent produces an answer to the query conditioned on the search and browsing trajectory.

To gather trajectories for both Chinese and English queries, we employ agents with DeepSeek-V3 (Liu et al., 2024) and Gemini-2.5-Flash (Gemini, 2025) as backbones. We then apply a two-stage filtering process to identify trajectories that truly accomplish the task. In the first stage, we use LLMs to compare the agent's answer with the golden reference answer following the setting of InfoDeepSeek, discarding trajectories where the outputs are inconsistent. In the second stage, to mitigate hallucination and the influence of an LLM's internal knowledge, we employ human evaluators to verify whether the correct answer can indeed be derived from the search and browsing trajectory itself.

Here is an example for trajectory:

---

**Example 2: Example for Trajectory**

QUERY: At which agrarian university did the president elected in Abkhazia in 2025 study?
TRAJECTORY:

```
[
    {
        "planning": "Search for the president elected in
        Abkhazia in 2025 and their educational background",
        "execution": {
            "name": "web_search",
            "args":
                {
                    "text": "President elected in Abkhazia
                    ↪  2025 university education"
                }
        },
        "observation": "title: CEC: Badra Gunba elected
            President of Abkhazia
        body: Badra Gunba was elected President of Abkhazia,
        ↪  and in the second round of presidential
        ↪  elections, the politician won 54.73\% of the
        ↪  vote. This was announced by the chairman of the
        ↪  Central Election Committee (CEC) Dmitry Marshan.
        url: https://abkhazia.news-pravda.com/en/world/
        2025/03/02/3676.html
        title: The new president of Abkhazia has been
        ↪  determined: Badra Gunba has won ...
        body: The Central Election Commission of Abkhazia has
        ↪  approved the results of the early presidential
        ↪  elections, which were won by Badra Gunba, who won
        ↪  almost 55\% of the vote. His opponent, Adgur
        ↪  Ardzinba, received about 42\% of the vote.
        url: https://abkhazia.news-pravda.com/en/world/
        2025/03/03/3786.html
```

---

```
            title: The 2025 presidential elections in Abkhazia: a
            ↪   turning point in ...
            body: On 1 March 2025, Abkhazia elected a new
            ↪   president, the 6 th since de facto independence
            ↪   in 1992. Over the years, power has shifted
            ↪   between position and opposition.
            url:
            ↪   https://defactostates.ut.ee/the-2025-presidential
            -elections-in-abkhazia-a-turning-point-in-abkhazian
            -russian-relations/
            title: Badra Gunba was elected President of Abkhazia
            ↪   ...
            body: Badra Gunba has been elected the new president
            ↪   of Abkhazia according to preliminary results of
            ↪   the republic's Central Election Commission. Gunba
            ↪   received 54.73% of votes in the second round of
            ↪   the presidential election in Abkhazia, while
            ↪   opposition leader Adgur Ardzinba, 41.54%, the
            ↪   commission specified.
            url: https://news-pravda.com/world/2025/
            03/02/1104147.html
            title: Explainer | Who's who in Abkhazia's 2025
            ↪   presidential elections
            body: Following the political crisis in Abkhazia in
            ↪   November 2024 that culminated in the ousting of
            ↪   former President Aslan Bzhaniya, new snap
            ↪   elections were scheduled for February 2025.
            url: https://oc-media.org/explainer-whos-who-in-
            abkhazias-2025-presidential-elections/"
        },
        {
            "planning": "Search for Badra Gunba's educational
            ↪   background or university attended",
            "execution": {
                "name": "web_search",
                "args": {
                    "text": "Badra Gunba university education"
                    }
                },
            "observation": "title: Biography of Badra Gunba –
            ↪   Pravda Abkhazia
            body: Badra Zurabovich Gunba was born on August
            ↪   14, 1981 in the city of Sukhumi in the
            ↪   Abkhazian ASSR (now Sukhum, Republic of
            ↪   Abkhazia). In 1998, he entered the Faculty of
            ↪   Economics of the Abkhazian State University.
            url: https://abkhazia.news-pravda.com/en/world/
            2025/03/02/3685.html
            title: Biography of presidential candidate Badra
            ↪   Gunba – Pravda Abkhazia
            body: Presidential candidate of the Republic of
            ↪   Abkhazia Badra Gunba was born on August 14,
            ↪   1981 in the city of Sukhumi in the Abkhazian
            ↪   ASSR. In 1998 , he graduated from Secondary
            ↪   school No. 20 in Sukhumi and entered the
            ↪   Economics Department of the Abkhazian State
            ↪   University.
```

```
                    url: https://abkhazia.news-pravda.com/en/world/
                    2025/01/13/454.html
                    title: Democracy Research Institute
                    body: Badra Gunba was born on August 14, 1981 in
                    ↪  S o khumi. His education and a significant
                    ↪  part of his career are connected with Russia.
                    ↪  In 2003, he graduated from the Faculty of
                    ↪  Economics of the N.I. Saratov State Vavilov
                    ↪  Agrarian University, defended his thesis in
                    ↪  2007 and received the title of Candidate of
                    ↪  Economic Sciences.
                    url: https://www.democracyresearch.org/eng/1582/
                    title: Badra Gunba: A New Leader or Bzhania 2.0?
                    ↪  - abkhazworld.com
                    body: The Central Election Commission (CEC) of
                    ↪  Abkhazia has announced that Badra Gunba has
                    ↪  won the presidential runoff election,
                    ↪  securing 54.73\% of the vote. His opponent,
                    ↪  Adgur Ardzinba, received 41.54\%, according
                    ↪  to preliminary results.
                    url:
                    ↪  https://abkhazworld.com/aw/opinion-editorial/
                    2988-badra-gunba-a-new-leader-or-bzhania-2-0
                    title: Badra Gunba Nominated for Abkhazia's
                    ↪  Presidential Election
                    body: Badra Gunba was born on 14 August 1981 in
                    ↪  Sukhum. He graduated in 2003 from the Faculty
                    ↪  of Economics at Saratov State Agrarian
                    ↪  University (now Saratov State University of
                    ↪  Genetics, Biotechnology, and Engineering),
                    ↪  specialising in Accounting and Auditing.
                    url:
                    ↪  https://www.abkhazworld.com/aw/current-affairs/
                    2914-badra-gunba-nominated-for-abkhazia-s-
                    presidential-election"
        },
        {
            "planning": "Terminate planning as the query is
            ↪  answered.",
            "execution": {
                "name": "task_complete",
                "args": {}
                }
        }
    ]
```

### D.1.2 PLANNING ABILITIES

Closed-ended questions in our setting are typically multi-hop problems. For such problems, the overall task planning and decomposition are not inherently difficult. The true challenge arises during execution, where the agent may encounter various contingencies, for example, failing to retrieve relevant content for a search query or obtaining contradictory evidence from different sources. In these cases, the model must dynamically adjust its plan in response to the external feedback.

Therefore, we mainly focused on stepwise planning, formulating questions that assess the agent's decision-making at each individual step. Even though there are some successful trajectories for a given task, there are often multiple plausible next steps at the local level (e.g., searching with semantically similar keywords). To capture this, we adopt a multiple-choice format: given the task

description and the first $T$ steps of the trajectory, the model must select the most reasonable next step, including both the plan and the corresponding action, from a set of candidate options. We include both plan and action explicitly in the options, since the plan essentially determines the tool to be invoked along with its parameters. The correct option corresponds to the $(T+1)$-th plan-action pair in the ground-truth trajectory, while incorrect options are generated by systematically degrading this ground-truth answer with LLMs.

To ensure the incorrect choices are meaningful yet clearly suboptimal, we design distinct degradation strategies depending on the type of tool call (search, browse, or terminate). This helps guarantee that the incorrect options deviate from the correct one in realistic but identifiable ways.

- When the next step is a **search** action, distractor options may include: (1) browsing a document already seen in the trajectory or irrelevant to solving the task; (2) searching for the next subtask before the current subtask has been resolved; (3) terminating the task; (4) issuing a redundant search even though the current tool invocation already suffices to solve the subtask; (5) producing a correct subtask with mismatched tool parameters (e.g., an inconsistent keyword, URL, or query that does not correspond to the subtask); (6) producing a correct subtask but invoking the wrong tool or malformed tool calls (e.g., similar but incorrect tool names or parameter names, type mismatches, or extraneous parameters).

- When the next step is a browse action, distractors may include: (1) browsing a document unrelated to the task; (2) skipping browsing and instead issuing a new search for the next subtask; (3) terminating the task; (4) repeating a previous search from the trajectory; (5) generating a correct subtask but mismatched parameters (e.g., incorrect keywords or URLs not aligned with the subtask); (6) generating a correct subtask but with erroneous tool invocation (e.g., incorrect tool names, parameter mismatches, type errors, or redundant parameters).

- When the next step is to terminate, distractors may include: (1) continuing to browse documents; (2) repeating the previous search or generating a new subtask-related search; (3) generating a correct subtask but with incorrect tool invocation (e.g., incorrect or inconsistent tool names/parameters, type errors, or additional irrelevant parameters).

For each of the three tool types, we define dedicated prompting strategies (see Prompt 3, 4, and 5). At each step of a successful trajectory, we generate one correct option and five incorrect options using the procedures above. The incorrect choices are then shuffled together with the correct answer to form the final multiple-choice question. This ensures that every step in a trajectory is converted into a test item that evaluates the model's ability to make sound stepwise planning decisions under uncertainty.

---

**Prompt 3: Incorrect Choices Generation for Search**

You are a behavior trajectory analysis assistant who can understand the task and the behavior trajectory of the intelligent agent very well, and judge what should and should not be done next. Based on the question and answer the agent needs to solve, the agent's trajectory, and its next action, you can understands the agent's behavior and generates five incorrect action options that the agent should not take next. Please follow the above instructions.

1. The agent's trajectory primarily includes search and browse behaviors, where each data point represents an action (a subtask ("task_name") and a tool call ("command") for the subtask) and its result ("result").

2. The last data point in the agent's trajectory contains the current subtask, the current tool call, and its result. The next action is the agent's next search behavior, including the subtask and the tool call for the subtask, but not the result.

3. Tool calls only include: {tool_set}.

4. Types of incorrect behaviors that may be considered include but are not limited to: (1) Browsing documents that appear in the trajectory but are not related to solving the problem; (2) Searching for the next subtask when the current tool call does not solve the current subtask (the correctness of the information obtained in the result can be determined based on the answer); (3) Ending the task; (4) The current tool call can

solve the current subtask but still performs a similar search (only the search terms are slightly changed); (5) The subtask is correct but does not correspond to the tool call parameters, such as the search keyword or browsed URL or question cannot correspond to the subtask (but still requires to be related to the problem to be solved by the agent); (6) The subtask is correct but the tool call is incorrect, such as similar but inconsistent tool names and parameter names, parameter type errors, and extra parameters (but the format must be parsable).

5. The next action can serve as a reference for the correct action, but it may not be the only correct one. Error action options should not include the next action, and these other possible correct actions should be avoided as much as possible. For example, searching for keywords similar to the next action (only with different wording).

6. The subtask in the error action option refers to the subtask that the agent will complete next. Do not include a description of how the error occurred. For example, instead of directly saying "browsing irrelevant documents," specify the specific document. Instead of saying "using general/repetitive search terms," say "continue adjusting search terms." Instead of saying "ending the task prematurely," say "ending the task." Avoid mentioning "using the wrong tool or parameters," "imprecise keywords," "no clear answer found," or "insufficient information."

7. Each incorrect action option should refer to the format of the next action, and the final output should be a JSON list.

Question: {query}
Answer: {answer}
Trajectory: {trajectory}
Next action: {action}
Based on the above requirements, generate five incorrect action options in the format of a JSON list:

```
[{
    "task_name": "Subtask Description 1",
    "command":{
        "name":"command name",
        "args":{
            "arg name":"value"
        }
    }
},
{
    "task_name": "Subtask Description 2",
    "command":{
        "name":"command name",
        "args":{
            "arg name":"value"
        }
    }
}]
```

Must be returned as a list to ensure that the task can be parsed by Python's json.loads function. Generate five incorrect action options:

---

**Prompt 4: Incorrect Choices Generation for Browsing**

You are a behavior trajectory analysis assistant who can understand the task and the behavior trajectory of the intelligent agent very well, and judge what should and should not be done next. Based on the question and answer the agent needs to solve, the agent's trajectory, and its next action, you can understands the agent's behavior and generates five incorrect action options that the agent should not take next.

1. The agent's trajectory primarily includes search and browse behaviors, where each data point represents an action (a subtask ("task_name") and a tool call ("command") for the subtask) and its result ("result").

2. The last data point in the agent's trajectory contains the current subtask, the current tool call, and its result. The next action is the agent's next browsing behavior, including the subtask and the tool call for the subtask, but not the result.

3. Tool calls only include: {tool_set}.

4. Types of incorrect behaviors that may be considered include but are not limited to: (1) Browsing documents that appear in the trajectory but are not related to solving the problem; (2) Directly searching for the next subtask without browsing the document; (3) Directly ending the task; (4) Repeating the search behavior in the previous trajectory (but with different wording); (5) The subtask is correct but does not correspond to the tool call parameters, such as the search keyword or browsed URL or question cannot correspond to the subtask (but still requires to be related to the problem to be solved by the agent); (6) The subtask is correct but the tool call is incorrect, such as similar but inconsistent tool names and parameter names, parameter type errors, and extra parameters (but the format must be parsable).

5. The next action can serve as a reference for the correct action, but it may not be the only correct one. Error action options should not include the next action, and these other possible correct actions should be avoided as much as possible. For example, searching for keywords similar to the next action (only with different wording).

6. The subtask in the error action option refers to the subtask that the agent will complete next. Do not include a description of how the error occurred. For example, instead of directly saying "browsing irrelevant documents," specify the specific document. Instead of saying "using general/repetitive search terms," say "continue adjusting search terms." Instead of saying "ending the task prematurely," say "ending the task." Avoid mentioning "using the wrong tool or parameters," "imprecise keywords," "no clear answer found," or "insufficient information."

7. Each incorrect action option should refer to the format of the next action, and the final output should be a JSON list.

Question: {query}
Answer: {answer}
Trajectory: {trajectory}
Next action: {action}
Based on the above requirements, generate five incorrect action options in the format of a JSON list:

```
[{
    "task_name": "Subtask Description 1",
    "command":{
        "name":"command name",
        "args":{
            "arg name":"value"
        }
    }
},
{
    "task_name": "Subtask Description 2",
    "command":{
        "name":"command name",
        "args":{
            "arg name":"value"
        }
    }
}]
```

Must be returned as a list to ensure that the task can be parsed by Python's json.loads function. Generate five incorrect action options:

---

**Prompt 5: Incorrect Choices Generation for Termination**

You are a behavior trajectory analysis assistant who can understand the task and the behavior trajectory of the intelligent agent very well, and judge what should and should not be done next. Based on the question and answer the agent needs to solve, the agent's trajectory, and its next action, you can understands the agent's behavior and generates five incorrect action options that the agent should not take next.

1. The agent's trajectory primarily includes search and browse behaviors, where each data point represents an action (a subtask ("task_name") and a tool call ("command") for the subtask) and its result ("result").

2. The last data point in the agent's trajectory contains the current subtask, the current tool call, and its result. The next action is the agent's next browsing behavior, including the subtask and the tool call for the subtask, but not the result.

3. Tool calls only include: {tool_set}.

4. Types of incorrect behaviors that may be considered include but are not limited to: (1) Continue browsing the document; (2) Continue searching the previous subtask or generate a new subtask related to the problem for searching; (3) The subtask is correct, but the tool call is incorrect, such as similar but inconsistent tool names and parameter names, incorrect parameter types, and extra parameters (but the format must be parseable). The 5 error behavior options must cover the above error types.

5. The next behavior can be used as a reference for the correct behavior. The error behavior options cannot include the next behavior.

6. The subtask in the error action option refers to the subtask that the agent will complete next. Do not include a description of how the error occurred. For example, instead of directly saying "browsing irrelevant documents," specify the specific document. Instead of saying "using general/repetitive search terms," say "continue adjusting search terms." Instead of saying "ending the task prematurely," say "ending the task." Avoid mentioning "using the wrong tool or parameters," "imprecise keywords," "no clear answer found," or "insufficient information."

7. Each incorrect action option should refer to the format of the next action, and the final output should be a JSON list.

Question: {query}
Answer: {answer}
Trajectory: {trajectory}
Next action: {action}
Based on the above requirements, generate five incorrect action options in the format of a JSON list:

```
[{
    "task_name": "Subtask Description 1",
    "command":{
        "name":"command name",
        "args":{
            "arg name":"value"
        }
    }
},
{
    "task_name": "Subtask Description 2",
    "command":{
        "name":"command name",
        "args":{
```

```
                "arg name":"value"
            }
        }
}]
```

Must be returned as a list to ensure that the task can be parsed by Python's json.loads function. Generate five incorrect action options:

### D.1.3 ACTION ABILITIES

In this setting, the agent's actions mainly consist of stepwise tool invocations and the final answer generation. Since tool use has already been evaluated in the above stepwise planning tasks, and once the plan is fixed, tool calls leave relatively little room for variation—this component is comparatively straightforward. Therefore, our primary focus here is on the task of producing the final answer based on the user query and the accumulated search and browsing trajectory.

For closed-ended questions, the answers are typically concise and well-defined, making them well suited to a text completion format. However, some of the original answers in InfoDeepSeek tend to be verbose. For example, multi-hop questions may include intermediate reasoning steps in the provided answer. To streamline this, we employ LLMs to shorten and standardize the answers by retaining only the answer to the final hop and enforcing a consistent format (e.g., for dates and numbers). Answers that remain overly long after shortening, or those involving multiple entities, are excluded. This refinement process is followed by manual validation, ensuring that the resulting answers are both accurate and concise. By doing so, we reduce the difficulty of completion for base models while preserving the essential correctness of the answers.

## D.2 OPEN-ENDED QUESTION

### D.2.1 TASK AND TRAJECTORY COLLECTION

We primarily collected open-ended questions from existing benchmarks such as DeepResearch-Bench (Du et al., 2025a) and Researchy Question (Rosset et al., 2024). Researchy Question contains English queries and a standard task decomposition plan. DeepResearch-Bench includes user queries in both Chinese and English, along with reference reports. For the questions in DeepResearch-Bench, we used high-performing Deep Research Agents (such as Claude Deep Research and Doubao Deep Research) to generate reports. We did not collect specific internal trajectories because these Deep Research Agents are commercial applications, and only the final results are accessible, making it difficult to obtain the intermediate trajectories.

### D.2.2 PLANNING ABILITIES

For planning abilities, we mainly focus on overall task planning. This is because information seeking for open-ended questions involves multiple aspects and tests the model's ability to break down and organize tasks at a high level. In contrast, specific step-by-step planning is not as challenging for open-ended questions as it is for closed-ended ones (because the answers are relatively easier to find). Overall planning also does not have a single optimal solution, so we adopt a multiple-choice format, where the model must select the best plan based on the current query.

We randomly sampled 300 questions from the Researchy Question test set that contained more than five plan steps and included both high-level and low-level plans (see Example 3 for more details). We use the standard plan in Researchy Question as the correct answer, and then degraded this plan to generate incorrect options. The methods for generating incorrect answers include:

- Randomly swapping two high-level plans and randomly scrambling low-level plans within a high-level plan, producing two incorrect options. This leads to logical inconsistencies in the report plan, such as presenting the application of a technology before explaining its definition. The combination of these two disruptions ensures errors.

- Swapping some low-level plans between two high-level plans, generating two incorrect options, which causes noticeable content misalignment.

- Randomly deleting some low-level plans, generating one incorrect option, which results in incomplete content.
- Randomly adding extra low-level plans, generating one incorrect option, which introduces irrelevant content.

Since the rules for error generation are simple and clear, the incorrect answers are automatically generated using these rules. These incorrect answers are then randomly mixed with the correct answers to create the final options.

---

**Example 3: Example for Researchy Question**

QUERY: why is so much money being printed
PLAN:
1. What does it mean to print money?
- How is money created and circulated in the economy?
- What are the different types of money and how are they measured?
2. How much money is being printed and by whom?
- What are the sources of data on money supply and growth?
- What are the roles and responsibilities of central banks and governments in money creation and management?
3. Why is money being printed and for what purposes?
- What are the economic objectives and challenges that motivate money printing?
- What are the monetary policy tools and instruments used to print money and influence interest rates and inflation?
- What are the fiscal policy measures and spending programs that are financed by money printing and borrowing?
4. What are the effects and consequences of money printing?
- How does money printing affect the value of money and exchange rates?
- How does money printing affect inflation and deflation?
- How does money printing affect economic growth and output?
- How does money printing affect income and wealth distribution and inequality?
- How does money printing affect debt and fiscal sustainability?
5. What are the alternatives and trade-offs to money printing?
- What are the costs and benefits of money printing compared to other policy options?
- What are the risks and uncertainties associated with money printing?
- What are the best practices and lessons learned from historical and international experiences of money printing?

---

### D.2.3 ACTION ABILITIES

For action capabilities, we primarily focus on the model's ability to generate reports, as the intermediate tool-use process is difficult to measure. Since base models often struggle with instruction-following to produce a long, cohesive report, we evaluate this ability using a multiple-choice format. The model is required to select the best report for a given query from a set of options. The correct answer is a high-quality report that we collected. To create the incorrect options, we systematically degrade this reference report by introducing flaws in its accuracy, logic, readability, and alignment with user's keypoints, ensuring the degraded versions are similar in length to the original so length cannot be used to determine the correct answer. Here are the specific types of flaws we introduce to create the distractors:

- Accuracy Issues: This includes: (1) replacing specific data with vague terms (e.g., "a lot," "some"); (2) making previously clear criteria ambiguous; and (3) presenting conclusions, explanations, or definitions that are vague or ambivalent. Please refer to Prompt 6 for more details.
- Logical Issues: This involves: (1) reversing cause and effect; (2) creating logical contradictions; (3) introducing incomplete arguments (e.g., overgeneralization or circular reasoning); and (4) adding extra arguments that are irrelevant to the conclusion. Please refer to Prompt 7 for more details.

- Readability Issues: This covers flaws such as: (1) disorganized formatting and paragraphing in two random sections; (2) grammatical errors and typos in two random sections; and (3) unclear explanations of professional terminology. Please refer to Prompt 8 for more details.

- Lack of Key Points: This involves degrading a report so it fails to meet the user's key requirements as defined by a benchmark-specific rubric. For example, if a user asks for an analysis of both the current state and future developments, the degraded report might only focus on the current state. Please refer to Prompt 9 for more details.

To generate the incorrect options, we randomly select three of the four flaw categories to create degraded reports, which are then mixed with the correct option. Since each option is a full report, these questions are typically quite long.

---

**Prompt 6: Incorrect Choices Generation for Accuracy Issues**

Please modify the reference report to generate a flawed report with accuracy issues.

1. The majority of the content in the flaw report should be consistent with the reference report, with only minor discrepancies.

2. Possible flaws include but are not limited to: (1) Replacing specific data with vague terms (such as "many" or "some"); (2) Making previously clear criteria unclear; (3) Conclusions, explanations, or definitions are too vague or ambiguous

3. The flaw report must contain at least two or more flaws, combining the various flaw types listed above.

4. The length of the flaw report should be the same as the reference report. If adding a flaw reduces the length of the report, add other content to make the flaw report the same length as the reference report.

5. Provide a detailed explanation of the flaw before generating the flaw report. The final output should be in the JSON list format given below.

Reference Report: article
Based on the above requirements, generate a flawed report with accuracy issues and a detailed explanation of the flaws in JSON list format:

```
[{
    "explanation": "Detailed explanation of the flaws in
    the report",
    "report": "Flawed Report"
}]
```

Must be returned as a list without incorrectly escaped backslashes to ensure that the above JSON list can be parsed by Python's json.loads function. Next, generate the flawed report and its explanation:

---

**Prompt 7: Incorrect Choices Generation for Logical Issues**

Please modify the reference report to generate a flawed report with logical issues.

1. The majority of the content in the flaw report should be consistent with the reference report, with only minor discrepancies.

2. Possible flaws include but are not limited to: (1) Reversing cause and effect; (2) Logical contradictions; (3) Incomplete arguments, such as overgeneralization or circular reasoning; (4) Adding extra evidence that is unrelated to the conclusion.

3. The flaw report must contain at least two or more flaws, combining the various flaw types listed above.

4. The length of the flaw report should be the same as the reference report. If adding a flaw reduces the length of the report, add other content to make the flaw report the same length as the reference report.

5. Provide a detailed explanation of the flaw before generating the flaw report. The final output should be in the JSON list format given below.

Reference Report: article
Based on the above requirements, generate a flawed report with logical issues and a detailed explanation of the flaws in JSON list format:

```
[{
    "explanation": "Detailed explanation of the flaws in
    the report",
    "report": "Flawed Report"
}]
```

Must be returned as a list without incorrectly escaped backslashes to ensure that the above JSON list can be parsed by Python's json.loads function. Next, generate the flawed report and its explanation:

---

**Prompt 8: Incorrect Choices Generation for Readability Issues**

Please modify the reference report to generate a flawed report with readability issues.

1. The majority of the content in the flaw report should be consistent with the reference report, with only minor discrepancies.

2. Possible flaws include but are not limited to: (1) Confusing layout and paragraphing in three random paragraphs in the middle section; (2) Grammatical errors and typos in three random paragraphs in the middle section; (3) Some technical terms are unclear.

3. The flaw report must contain at least two or more flaws, combining the various flaw types listed above.

4. The length of the flaw report should be the same as the reference report. If adding a flaw reduces the length of the report, add other content to make the flaw report the same length as the reference report.

5. Provide a detailed explanation of the flaw before generating the flaw report. The final output should be in the JSON list format given below.

Reference Report: article
Based on the above requirements, generate a flawed report with readability issues and a detailed explanation of the flaws in JSON list format:

```
[{
    "explanation": "Detailed explanation of the flaws in
    the report",
    "report": "Flawed Report"
}]
```

Must be returned as a list without incorrectly escaped backslashes to ensure that the above JSON list can be parsed by Python's json.loads function. Next, generate the flawed report and its explanation:

---

**Prompt 9: Incorrect Choices Generation for Missing Keypoints**

Please modify the reference report to generate a flawed report with missing keypoints.

1. The issues addressed in the flawed report must be consistent with those in the reference report; significant differences in the specific content are permitted.

2. The generated flawed report must not meet all the keypoints provided. If the report itself does not meet all the key points, then narrow the breadth or angle of the report coverage to miss more key points.

3. Modifying some keypoints may involve significant deletions and revisions; you may add other content to maintain a similar length with reference report.

4. The length of the flawed report must be the same as the reference report; length alone cannot be used to distinguish between the reference report and the flawed report.

5. Provide a detailed explanation of the flaw before generating the flaw report. The final output should be in the JSON list format given below.

Reference Report: article
Keypoints: keypoint
Based on the above requirements, generate a flawed report with missing keypoints and a detailed explanation of the flaws in JSON list format:

```
[{
    "explanation": "Detailed explanation of the flaws in
    the report",
    "report": "Flawed Report"
}]
```

Must be returned as a list without incorrectly escaped backslashes to ensure that the above JSON list can be parsed by Python's json.loads function. Next, generate the flawed report and its explanation:

### D.2.4 ATOMIC ABILITIES

For atomic abilities, our focus is on a key characteristic of deep research: a model's capacity to cite relevant sources to support its statements. While open-ended questions don't have a single correct answer, it's still crucial for the response to be factually grounded. Therefore, the model's ability to correctly identify which information sources support its claims is very important. To evaluate this, we use a multiple-choice format. Given a report and a specific webpage cited within it, we present several statements from the report and require the model to identify which are supported by the content of the webpage.

Following DeepResearch Bench, we use LLMs to extract all cited statements from a report and the webpages they reference. All statements from the report that cite a specific webpage are grouped together to form the correct answer. To create the distractors, we use an LLM to generate several statements that are not supported by the webpage's content. These are then combined with the correct statements to form a set of six choices. While a correct answer may contain multiple statements that cite the same webpage, we limit the number to a maximum of three. For citation extraction and web scraping, we follow DeepResearch Bench and the process for generating incorrect options is detailed in Prompt 10 Due to the length of the reports and webpage content, these questions are typically quite long.

---

**Prompt 10: Incorrect Choices Generation for Citation**

You are a professional question maker, and your task is to generate incorrect distractor options for a multiple-choice question based on the provided article and web page content. Given an article, a webpage referenced by the article, and some statements in the article that is supported by the webpage, generate {num} false statements.

1. Each statements is a single, concise sentence.

2. The false statement must be a sentence from the article that is unsupported by the webpage.

3. The subject of the false statement must be mentioned in the webpage and cannot be completely absent from the webpage.

4. Possible false statements include but are not limited to: (1) A statement in the article that refers to a subject mentioned in the webpage, but the webpage does not support the statement; (2) Modifying a statement in the article that is supported by the webpage to make it unsupported.

5. When generating false statements, avoid the given supported statements and similar ones. Other supported statements in the article may also exist, and these should be avoided when generating false statements.

6. Each false statement should provide an explanation, which will be output in the following JSON list format.

Article: {article}
This is a webpage cited by the article: {webpage_content}
The following are some statements in the article that are supported by the webpage (please do not duplicate these statements): {statements}
Based on the above requirements, generate {num} unsupported, false statements in JSON list format:

```
[{
    "statement": "False statement 1",
    "explanation": "Explain why this statement is wrong
    and unsupported",
},
{
    "statement": "False statement 2",
    "explanation": "Explain why this statement is wrong
    and unsupported",
}]
```

Must be returned as a list to ensure that the task can be parsed by Python's json.loads function. Next, generate {num} false statements:

### D.3 STATISTICS

We have compiled statistics on the number of different types of questions in deep research scenarios, as shown in Table 5. The closed-end questions are mainly drawn from the InfoDeepSeek benchmark, which contains both Chinese and English queries; accordingly, our questions are provided in both languages. The planning component of the open-end questions is based on Researchy Question, which is only available in English, so our corresponding questions are also English-only. In addition, the planning component of the open-end questions is derived from DeepResearch Bench, which includes both Chinese and English samples, and we have constructed questions in both languages for this case as well.

Table 5: The statistics for APTBench-DR (Deep Research)

|  | Closed-ended Question | | Open-ended Question | | |
| --- | --- | --- | --- | --- | --- |
|  | Planning | Action | Planning | Action | Citation |
| English | 614 | 212 | 298 | 111 | 173 |
| Chinese | 417 | 138 | / | 103 | 189 |
| Total | 1031 | 350 | 298 | 214 | 362 |

## E EXPERIMENT DETAILS

### E.1 EXPERIMENT SETUP

We use the corresponding default configurations of the tested open-sourced models to conduct the experiments. The greedy decoding strategy is utilized to minimize randomness. We use vLLM Kwon et al. (2023) with FlashAttention2 Dao et al. (2022) as the inference engine. If the input

length is larger than the model's max sequence length, we will truncate the input from the head and tail part.

## E.2 EVALUATION PROMPT AND EXAMPLES FOR ENVSETUP

**Questions for Planning Abilities**   For evaluating planning ability, the questions require model to select the best overall environment setup plan based on repository information. We used 3-shot prompting for evaluation, with the specific prompt provided in Prompt 11. As the data samples in this part are generally lengthy, we did not include an example here; interested readers may refer to our dataset.

---

**Prompt 11: Evaluation Prompt for Planing in EnvSetup (3-shot)**

Request: Choose the correct execution plan to setup the environment for running the repository according to the repository information.
Repo information: {DEMONSTRATION_REPO_INFO}
Execution plans: {DEMONSTRATION_CHOICES}
The correct execution plan is ({DEMONSTRATION_ANSWER})
...
Request: Choose the correct execution plan to setup the environment for running the repository according to the repository information.
Repo information: {REPO_INFO}
Execution plans: {CHOICES}
The correct execution plan is (

---

**Questions for Action Abilities.**   For evaluating action ability, the questions require model to predict the next command given excution plan and the previously executed commands in the plan. We also used 3-shot prompting for evaluation, with the specific prompt provided in Prompt 12 and a corresponding data sample shown in Example 4.

---

**Prompt 12: Evaluation Prompt for Action in EnvSetup (3-shot)**

Request: Write the next execution command given the execution plan and the previously executed commands .
Execution plan: {DEMONSTRATION_EXE_PLAN}
Previous executed commands: {DEMONSTRATION_EXE_CMDS}
The next command:

```bash
{DEMONSTRATION_ANSWER}
```

...
Request: Write the next execution command given the execution plan and the previously executed commands . Execution plan: {EXE_PLAN}
Previous executed commands: {EXE_CMDS}
The next command:

```bash
```

---

**Example 4: Example for Action in EnvSetup**

EXE_PLAN: step1: Set up the environment using Anaconda and install required dependencies;
step2: Download pretrained models and FID reference sets;
step3: Generate training data for LD3 using the teacher solver for CIFAR-10;
step4: Generate training data for LD3 using the teacher solver for Stable Diffusion;
step5: Train LD3 on the generated CIFAR-10 training data;

---

```
step6: Compute FID for Stable Diffusion using default timesteps;
step7: Compute FID for Stable Diffusion using custom timesteps.
EXE_CMDS:

    conda env create -f requirements.yml
    conda activate ld3
    pip install -e ./src/clip/
    pip install -e ./src/taming-transformers/
    pip install omegaconf
    pip install PyYAML

ANSWER: pip install requests
```

**Questions for ErrorHandle Abilities.** Exception-handling is the core atomic ability we focus on in EnvSetup. The model is required to select the best execution plan to fix an issue based on the exception description. This ability is evaluated using 3-shot prompting, with the specific prompt provided in Prompt 13.

---

**Prompt 13: Evaluation Prompt for ErrorHandle in EnvSetup (3-shot)**

Request: The environment is setup according to the repo setup steps, but an issue occurred. Choose the correct execution plan to fix the issue.
Repo setup: {DEMONSTRATION_SETUP}
Issue title: {DEMONSTRATION_ISSUE_TITLE}
Issue description: {DEMONSTRATION_ISSUE_BODY}
Execution plans: {DEMONSTRATION_CHOICES}
The correct execution plan is ({DEMONSTRATION_ANSWER})
...
Request: The environment is setup according to the repo setup steps, but an issue occurred. Choose the correct execution plan to fix the issue.
Repo setup: {SETUP}
Issue title: {ISSUE_TITLE}
Issue description: {ISSUE_BODY}
Execution plans: {CHOICES}
The correct execution plan is (

---

### E.3    EVALUATION PROMPT AND EXAMPLES FOR ISSUEFIX

**Questions for Planning and Action Abilities.** For evaluating planning ability in the IssueFix task, the questions require generating the next step of the plan based on the previous execution trajectory. This is evaluated using 3-shot prompting, with the specific prompt shown in Prompt 14. For evaluating action ability, the questions require writing the next command based on the previous trajectory, also evaluated using 3-shot prompting, with the specific prompt shown in Prompt 15. As the data samples for these tasks are generally lengthy, we did not include examples here; interested readers may refer to our dataset.

---

**Prompt 14: Evaluation Prompt for Planing in IssueFix (3-shot)**

Request: Select the correct next step plan given the previous executed trajectory.
Previous executed trajectory: {DEMONSTRATION_TRAJECTORY}
Choices: {DEMONSTRATION_CHOICES}
The correct next step plan is ({DEMONSTRATION_ANSWER})
...
Request: Select the correct next step plan given the previous executed trajectory.
Previous executed trajectory: {TRAJECTORY}
Choices: {CHOICES}
The correct next step plan is (

---

---

**Prompt 15: Evaluation Prompt for Action in IssueFix (3-shot)**

Request: Write the next execution command from the tool set according to the previous executed trajectory.
Tool set: {DEMONSTRATION_TOOLS}
Previous executed trajectory: {DEMONSTRATION_TRAJECTORY}
The next command:

```bash
{DEMONSTRATION\_ANSWER}
```

...
Request: Write the next execution command from the tool set according to the previous executed trajectory.
Tool set: {TOOLS}
Previous executed trajectory: {TRAJECTORY}
The next command:

```bash
```

---

**Questions for Atomic Abilities – Bug Localization.** Bug localization is an atomic ability under the IssueFix task. It requires the model to select the code snippet causing the bug based on the issue statement. It is evaluated using 3-shot prompting, with the specific prompt shown in Prompt 16.

---

**Prompt 16: Evaluation Prompt for Bug Localization in IssueFix (3-shot)**

Request: Select the code snippet generating the bug described in the issue statement.
Issue: {DEMONSTRATION_ISSUE}
Choices: {DEMONSTRATION_CHOICES}
The bug code snippet is ({DEMONSTRATION_ANSWER})
...
Request: Select the code snippet generating the bug described in the issue statement.
Issue: {ISSUE}
Choices: {CHOICES}
The bug code snippet is (

---

**Questions for Atomic Abilities – Fix Patch.** Fix Patch is also an atomic ability under the IssueFix task. It requires the model to select the correct patch to fix the bug based on the issue. It is evaluated using 3-shot prompting, with the specific prompt shown in Prompt 17. Since the data samples for this task are generally very long, we did not provide examples here; interested readers may refer to our dataset.

---

**Prompt 17: Evaluation Prompt for Fix Patch in IssueFix (3-shot)**

Request: Select the correct fix patch to the issue.
Issue: {DEMONSTRATION_ISSUE}
Choices: {DEMONSTRATION_CHOICES}
The correct patch to the issue is ({DEMONSTRATION_ANSWER})
...
Request: Select the correct fix patch to the issue.
Issue: {ISSUE}
Choices: {CHOICES}
The correct patch to the issue is (

---

**Questions for Atomic Abilities – Unit Test Generation.** Unit Test Generation is also an atomic ability under the IssueFix task. It requires the model to generate a unit test to reproduce the problem described in the issue. It is evaluated using 3-shot prompting, with the specific prompt shown in Prompt 18.

---

**Prompt 18: Evaluation Prompt for Unit Test Generation in IssueFix (3-shot)**

Question: Which is the correct test patch that can reproduce the issues described in the problem statement?
Problem statement: {DEMONSTRATION_PROBLEM}
Test patches: {DEMONSTRATION_CHOICES}
The answer is ({DEMONSTRATION_ANSWER})

...

Question: Which is the correct test patch that can reproduce the issues described in the problem statement?
Problem statement: {PROBLEM}
Test patches: {CHOICES}
The answer is (

---

### E.4 EVALUATION PROMPT AND EXAMPLES FOR CLOSED-ENDED QUESTIONS

**Questions for Planning Abilities.** Since base models often struggle to follow instructions and formatting constraints, we adopted few-shot prompting to ensure correct outputs. For the multiple-choice questions used to evaluate planning ability, we employed 3-shot prompting, with the evaluation prompt shown in Prompt 19.

---

**Prompt 19: Evaluation Prompt for Planing in Closed-ended Question (3-shot)**

Request: Given the question to be answered, the executed searching and browsing trajectory, and several possible next steps, determine which step is correct and most reasonable
Question: {DEMONSTRATION_QUERY}
Executed trajectory: {DEMONSTRATION_TRAJECTORY}
Tool set:

```
[
    {
        "name": "web_search",
        "description": "Perform an internet search.",
        "parameters": {
            "type": "object",
            "properties": {"text": {"type": "str",
            ↪  "description": "Search query."}}
            },
        "returns": {"description": "", "type": "str"},
        "required": ["text"]
    },
    {
        "name": "browse_website",
        "description": "Browse a specific website using the
        ↪  provided URL link. ",
        "parameters": {
            "type": "object",
            "properties": {
                "url": {"type": "str", "description": "The
                ↪  website's URL link."},
                "question": {"type": "str", "description":
                ↪  "The specific content or topic sought on
                ↪  the website."}}},
        "returns": {"description": "", "type": "str"},
        "required": ["url", "question"]
    },
    {
        "name": "task_complete",
```

---

```
            "description": "Indicate task completion without the
            ↪   need for further functions. ",
            "parameters": {"type": "object", "properties": {}},
            "returns": {"description": "", "type": ""},
            "required": []
        }
    ]
```
Choices of next step: {DEMONSTRATION_CHOICES}
The corrected next step should be ({DEMONSTRATION_ANSWER})

...

Request: Given the question to be answered, the executed searching and browsing trajectory, and several possible next steps, determine which step is correct and most reasonable
Question: {QUERY}
Executed trajectory: {TRAJECTORY}
Tool set: ... (The same as demonstration tool set)
Choices of next step: {CHOICES}
The corrected next step should be (

**Questions for Action Abilities.** For the multiple-choice questions used to evaluate action ability, we employed 3-shot prompting, with the evaluation prompt shown in Prompt 20 and corresponding data samples provided in Example 5. The trajectory part is omitted in the data sample because it is lengthy and has already been presented earlier (see Example 2).

---

**Prompt 20: Evaluation Prompt for Action in Closed-ended Question (3-shot)**

Request: Get the answer to the question from the searching and browsing trajectory
Question: {DEMONSTRATION_QUERY}
Searching and browsing trajectory: {DEMONSTRATION_TRAJECTORY}
The answer is [DEMONSTRATION_ANSWER}]

...

Request: Get the answer to the question from the searching and browsing trajectory
Question: {QUERY}
Searching and browsing trajectory: {TRAJECTORY}
The answer is [

---

**Example 5: Example for Planning in Closed-ended Question**

QUERY: At which agrarian university did the president elected in Abkhazia in 2025 study?
TRAJECTORY: ... (omitted as we present it previous in previous Example 2)
ANSWER: Saratov State Agrarian University

---

E.5  EVALUATION PROMPT AND EXAMPLES FOR OPEN-ENDED QUESTIONS

For the evaluation of planning, action, and citation abilities in open-ended questions, we also adopt a few-shot prompting setup, with the corresponding prompts provided in Prompt 21, 22, and 23. Specifically, planning and citation were evaluated with 3-shot prompting, while the action questions used 2-shot prompting. Since the data samples in this part are generally very long, readers can refer to our dataset for the cases.

---

**Prompt 21: Evaluation Prompt for Planning in Open-ended Question (3-shot)**

Request: Given a question, you need to research and write a report. Choose the best structure for your report from the options below
Question: {DEMONSTRATION_QUERY}
Choices of report structure: {DEMONSTRATION_CHOICES}

---

The best report structure is ({DEMONSTRATION_ANSWER})

...

Request: Given a question, you need to research and write a report. Choose the best structure for your report from the options below
Question: {QUERY}
Choices of report structure: {CHOICES}
The best report structure is (

---

**Prompt 22: Evaluation Prompt for Action in Open-ended Question (2-shot)**

Request: Below are four reports generated for the given problem. Please select the one that has best accuracy, readability and logic.
Problem: {DEMONSTRATION_QUERY}
Choices of reports: {DEMONSTRATION_CHOICES}
The best report is ({DEMONSTRATION_ANSWER})

...

Request: Below are four reports generated for the given problem. Please select the one that has best accuracy, readability and logic.
Problem: {QUERY}
Choices of reports: {CHOICES}
The best report is (

---

**Prompt 23: Evaluation Prompt for Citation in Open-ended Question (3-shot)**

Question: Which of the following statements appear in the article and are also strongly supported by the content of the webpage?
Article: {DEMONSTRATION_ARTICLE}
Web page: {DEMONSTRATION_WEB_PAGE}
Choices of statements: {DEMONSTRATION_CHOICES}
The supported statements are ({DEMONSTRATION_ANSWER})

...

Question: Which of the following statements appear in the article and are also strongly supported by the content of the webpage?
Article: {ARTICLE}
Web page: {WEB_PAGE}
Choices of statements: {CHOICES}
The supported statements are (

