# OpenReview forum: "APTBench: Benchmarking Agentic Potential of Base LLMs During Pre-Training"
_ICLR.cc/2026/Conference — Submitted to ICLR 2026_

### Official Review · Reviewer_Ggtk · 2025-10-29

**Soundness:** 1
**Presentation:** 2
**Contribution:** 2
**Rating:** 2
**Confidence:** 3

**Summary:**

This paper introduces APTBench, a new benchmark designed to evaluate the "agentic potential" of base LLMs during the pre-training stage.

The authors identify a gap where current pre-training benchmarks focus on static skills (like knowledge or reasoning) rather than agentic capabilities, while existing agent benchmarks are mostly end to end, requiring the model to be able to following multi-turn instructions and thus unsuitable for evaluating base models that only undergoes pre-training.

APTBench's main contribution is a novel methodology for converting agent trajectory logs generated from existing agent benchmark into multiple-choice or text completion questions that are more suitable for evaluating base models, and target specifically on the planning, action and atomic abilities essential for powering agents. The authors demonstrate the method by applying it to two different domains, coding and deep research. Results show that the score is closely correlated with final agentic capabilities of the model.

**Strengths:**

- The paper addresses a clear, important, and timely problem. As the field moves to integrate agent-specific data into pre-training, there is a critical need for a benchmark to measure "agentic potential" before the costly post-training stage. The paper correctly identifies this significant gap.
- Instead of just creating another end-to-end agent task, the work proposes a novel and creative framework to convert existing, complex agent trajectories into a static, multiple-choice/text-completion format. This makes evaluation on base models, which struggle with multi-turn interactions, feasible.

**Weaknesses:**

- The main weakness is around experiment design and results. The experimental results are mixed and inconclusive (e.g., as seen in Table 2, LLAMA3.2-3B has 14 for planning on EnvSetup, but 30 for IssueFix, LLAMA4 has 56 on EnvSetup and 50 on IssueFix, both seem to be good and bad  at planning depending on the split). This seem to suggest that the benchmark may be measuring domain-specific knowledge (e.g., knowledge of coding syntax, familiarity with research topics) rather than the intended general-purpose reasoning and planning abilities for agent. Similarly I don't think use of instruction tuned model results (in correlation analysis) is a fair comparison, as that is heavily impacted by the post-training, and not just base model potential.

- The benchmark's design of "atomic" and "general" actions feels overly specific to the domains of coding and research, rather than capturing general agentic behavior. True agentic potential involves more abstract capabilities, such as the ability to use new tools on the fly, adapt to unseen environments, or reason over novel information structures. The current tasks seem to equate "agentic potential" with "potential for this specific task," which deviates significantly from the paper's stated motivation.

- A primary contribution is presented as the method for converting datasets. However, this method appears tightly coupled to the specific structures of the two source datasets. This lack of generality limits the impact of the proposed methodology.

- The current benchmark appears to be derived from successful or "golden" trajectories. A large, non-trivial part of agentic intelligence is the ability to handle errors, recover from failed actions, and re-plan from unexpected states. By omitting error trajectories, the benchmark fails to evaluate a model's robustness and adaptability, which are critical components of true agentic potential.

**Questions:**

- The paper's core motivation is to evaluate base models, but the correlation analysis in Figure 1 and other experiments seem to use instruction-tuned models (e.g., Llama-3-Chat). Could you clarify on this? Since post-training heavily influences a model's ability to plan and follow instructions, how can these results be used to support claims about the base model's inherent potential?

- I found the results in Table 2 difficult to interpret as a clear signal of general planning. For instance, Llama-3.2 3B has a planning score of 14 on "EnvSetup" but 30 on "IssueFix," while Llama 4 is high on both (56 and 50). This high variance suggests the benchmark might be testing domain-specific knowledge for that task rather than a general, transferable planning ability. Could you provide an explanation for this variance and provide further support for the claim that this measures general planning? Also there seem to be outliers in result, e.g, GLM4.5 has 0 for action on IssueFix, why is that?

---

> ### Author Response · Authors · 2025-11-24
>
> Thank you very much for your careful reviews and constructive feedback. We address your concerns point by point below.
>
> **Q1**: Experiment results are mixed and inconclusive.
>
> **A1**: Since our benchmark is mainly designed for the pre‑training stage, we aim to observe the gradual change of a model’s agentic capabilities throughout pre‑training. As a result, **absolute scores can vary across different evaluation items, but the relative ranking needs to remain consistent**.
> For example, for the base models Llama‑3.2‑3B and Llama‑4 that you mentioned:
> Llama‑4 ranks 10/14 and 8/14 on EnvSetup and IssueFix, while Llama‑3.2‑3B ranks 14/14 and 13/14.
> In the average score ranking on APTBench‑SWE, Llama‑4 ranks 10/14 and Llama‑3.2 ranks 14/14, showing strong consistency in relative ordering.Even when switching domains from SWE to DR, the same pattern holds: on APTBench‑DR, Llama‑4 ranks 8/14 and Llama‑3.2 ranks 14/14, mirroring their positions on APTBench‑SWE.
>
>
> **Q2**: Why using instruct-tuned model in the correlation analysis.
>
> **A2**: The purpose of APTBench is to provide, during the pre‑training stage, an **evaluation proxy** that is highly correlated with a model’s agentic ability after post-train. Therefore, we use the performance of post‑trained models on several representative downstream agent benchmarks to examine whether APTBench correlates with them.
> It is true that post‑training has a strong impact on downstream metrics. However, applying a fully aligned post‑training procedure to all evaluated base models would be extremely costly and beyond our current compute budget. Moreover, we do not know how each model’s post‑training process is designed to best activate their model’s agentic capabilities. Thus, we can only evaluate the publicly released post‑trained models directly (and assume that their post‑training reasonably brings out the potential of their base models).
> We find that, **despite the differences in post‑training procedures, the final downstream results still show strong correlation with APTBench**, especially compared with other commonly used base‑model evaluation metrics. This is exactly what we aim for: APTBench can serve as a post‑training‑agnostic indicator of a base model’s agentic potential.
>
> **Q3**: The benchmark's design of "atomic" and "general" actions feels overly specific to the domains of coding and research.
>
> **A3**: Our approach is to start from the two most common scenarios of today's agent models, then decompose them into more abstract capability components: Planning & Action (Tool using). We believe these two aspects is the most fundamental ability of an agent foundation model. We further extend the evaluation dimensions with atomic skills (these are domain specific).
> The three capabilities you mentioned essentially concern generalization to new contexts (such as new tools, new environments, or new information structures), and indeed these are very important. **Our experimental results can partially reflect a base model’s ability to generalize to new contexts across scenarios**. For example, as shown in Figure 4(a), the correlation between APTBench‑DR and SWE‑Bench is higher than the correlation between EvalPlus and SWE‑Bench (Pearson's r=0.78 vs r=0.17), even though both of the latter are coding benchmarks. This indicates that our benchmark **captures more general agentic abilities rather than domain‑specific skills**.
>
> **Q4**: Benchmark construction method lacks generality.
>
> **A4**: A general agent trajectory in most tasks consists of planning and action steps. Therefore, we designed data‑generation schemes to evaluate these two core abilities: introducing incorrect options for planning, and text completion on the tool calling step for action. This generation framework is **independent and applicable of any specific scenario**.
>
> **Q5**: Only using successful trajectorise lacks of error handling.
>
> **A5**: Using successful trajectories does not mean we cannot evaluate a model’s ability to recover from failures, because **successful trajectories also contain segments where the model performs error reflection**. For example, SWE trajectories include re‑fix/reflection steps, and DR trajectories include re‑search behaviors. Our planning/action tests include these error recover steps of testing if the model could plan or act correctly given the error.
> The reason we use successful trajectories is that they allow us to more reliably obtain correct steps to serve as our ground truth plan/action. In contrast, for a failed trajectory, it is often unclear exactly where things went wrong. With our construction method, we can reliably ensure that each negative choice is strictly worse than its corresponding positive choice, which guarantees the correctness of the evaluation framework.

---

> > ### Comment · Reviewer_Ggtk · 2025-11-28
> >
> > Thank for the response and clarification. Some follow ups:
> >
> > R1:  Regarding the mixed result
> >
> > I agree that relative ranking is more important for benchmark, as after all, the purpose of the benchmark is to pick among model candidates which one is better. However, based on the result in Table 2:
> > - Llama3.2-3B is worse than Qwen3-1.7B in EnvSetup Plan, but better on IssueFix Plan
> > - Llama4 is worse than GLM on EnvSetup Plan, then better on IssueFix Plan
> > Also Table 2 is a bit hard to interpret with so many numbers. And then the given scores of different sub items are of different scale, the average of them might be affected and not well represented of the overall performance, e.g, if the gap between some item is large, or a model get extremely low score on some items.
> >
> > Another example if comparing DeepSeek-v3 and DeepSeek-v3.1. These two are very close on APTBench, or DeepSeek-v3.1 is slightly better on APTBench-SWE, and more improvement on APTBench-DR. However, checking the correlation analysis in Figure 1, DeepSeek-v3.1 is much better than DeepSeek-v3 on SWE-Bench and Terminal Bench, and v3.1 being slightly worse on $\tau$ bench. This might be due to the difference in post-training. But nevertheless it still renders the results a bit inconclusive
> >
> > R2: Regarding the usage of instruction tuned model for correlation analysis
> > I understand the limit of running full post-training for all models. But I feel to validate the contribution of the paper, some controlled experiments are still necessary, even if it is just on one of two small models, otherwise there are just too many variables mixed
> >
> > Given this, I would keep my current score and recommend revise the experiment design and results presentation.

---

### Official Review · Reviewer_Cm9U · 2025-10-30

**Soundness:** 3
**Presentation:** 4
**Contribution:** 3
**Rating:** 6
**Confidence:** 4

**Summary:**

This paper proposes a new benchmark, APTBench, that can be used to assess the agentic capabilities of pre-trained base models. Standard text benchmarks like MMLU, GSM8K, EvalPlus etc. are not good indicators for agentic tasks like software engineering (SWE-like) or Deep Research kind of setups and benchmarks like SWE-Bench, Terminal Bench, etc. The idea is to leverage multi-turn agentic traces and successful agent trajectories into single-turn MCQ format questions or text completion questions. And then these can be evaluated using standard metrics like exact-match and accuracy, etc. The paper shows correlations scores of base models on standard text benchmarks to be low with final post-trained model capabilities on agentic tasks, but APTBench on the other hand, exhibits higher correlation scores.

**Strengths:**

- The paper tackles an important problem of assessing base model capabilities for agentic purposes and real-world coding scenarios. A lot of post-trained benchmarks exist like SWE-Bench, MLE-Bench, Terminal-Bench, etc. but they can be only be evaluated on post-trained models due to their multi-turn and iterative setup. No such benchmarks exist for pre-trained models.
- Instead of just a benchmark, the authors detail the generation pipeline for building pre-training benchmarks for agentic capabilities, which can be used in the future to refresh/update the benchmark.
- The evaluation framework focuses on real-world use-cases like SWE and DeepResearch, which are representative of complex multi-step tasks an AI agent might tackle.
- High correlation of APTBench with downstream post-trained benchmarks shows the efficacy of measuring the potential of models during pre-training on agentic use-cases.

**Weaknesses:**

- I might've missed this but the paper does not mention clearly what models are used to perturb the correct ground truth to generate the incorrect options for the LLMs.
- How do different models affect the benchmark construction quality and correlation scores if different models are used to generate the traces/MCQ options. That might be an important indicator for the usability of this benchmark for pre-training.
- EM and Rouge scores are not reliable indicators of performance. I would like to see the correlation between negative log-likelihood (NLL) of the appended text completion vs the final downstream performance on tasks like SWE-Bench. Lower the NLL, higher the post-trained performance should be.
- Not really a weakness, but there are some typos here and there like GSM8K is incorrectly referenced as GPM8K at various places and opening quotes are not used correctly (" instead of `` in latex leading to quote mismatch).

**Questions:**

I have listed some of my questions in the weakness section above, but here are a couple more:

1. Can the authors show an equivalent figure similar to Figure 8 for APTBench (SWE and DR) vs Terminal-Bench and Tua as well. The main paper only shows the correlation b/w APTBench and SWE-Bench in Figure 4.
2. Instead of EM/Rouge scores, I would like to see LLM-as-a-judge and NLL comparison as well.

I would be happy to increase my score to 8 if the authors are able to address the weaknesses/questions.

---

> ### Author Response · Authors · 2025-11-24
>
> Thank you very much for the constructive feedback and the recognition of our contribution.
> We will address all your concerns in the following points:
>
> **Q1**: What models are used to generate the incorrect options and benchmark construction?
>
> **A1**: We use deepseek-v3-0324 and gemini-2.5-pro to generate the trajectories and the incorrect options for EnvSetup and APTBench-DR. For each question, we randomly select from these two models, trying to have better data diversity. For the IssueFix dataset, we use claude-4-sonnet to generate trajectories and incorrect options as Claude performs best on SWE tasks.
> We try to maintain a good diversity and use more closed source LLMs as data generator.
> In fact, different LLMs generate consistant MCQ options, because we use very specific error types to generate the wrong options. Appendix C.1.2 prompt 2 is one example, there is 3 types of error: dependency shuffle, step omission and step addition.
> We would like to thank you for poinint this out and these information will be updated in Appendix of the paper.
>
> **Q2**: Add LLM-as-a-judge and NLL comparison.
>
> **A2**: Using LLM-as-a-judge is a good suggestions to evaluate the text completion task. We list all the results of LLM-as-a-judge with original EM and Rouge scores in the updated Table2&3. We can find that LLM-as-a-judge has a higher score, because it will find more equivelant answers compared to exact match. Nevertheless, it shows similar relative ranking of models with the original EM/Rouge scores.
>
> As for the NLL comparision, we think it is a little bit hard to compare across different model families because they use different tokenizer and it will affect NLL. We test the Qwen3 family, it does show a clear pattern that better APTBench results with lower NLL.
>
> |  Base Model Names |      EnvSetup      |                    |     IssueFix     |                  | Close-end Question |                    |                    |                    |
> |:-----------------:|:------------------:|:------------------:|:----------------:|:----------------:|:------------------:|:------------------:|:------------------:|:------------------:|
> |                   | Action  (LLM,1084) | Action  (NLL,1084) | Action (LLM,241) | Action (NLL,241) | Summ_Ans  (en,LLM) | Summ_Ans  (en,NLL) | Summ_Ans  (zh,LLM) | Summ_Ans  (zh,NLL) |
> |  Qwen3-1.7B-Base  |       20.39        |        0.09        |      23.24       |       0.11       |        52.83       |        0.365       |        51.45       |       0.22       |
> |   Qwen3-4B-Base   |       33.58        |        0.07        |      25.73       |       0.10       |        55.19       |        0.307       |        60.87       |       0.16       |
> |   Qwen3-8B-Base   |       38.19        |        0.05        |      31.54       |       0.07       |        54.72       |        0.295       |        63.77       |        0.11        |
> | Qwen3-30BA3B-Base |       34.96        |        0.07        |      32.78       |       0.07       |        57.08       |       0.2127       |        61.59       |       0.07       |
>
> **Q3**: APTBench correlation with Tau2-Bench and Terminal Bench.
>
> **A3**: We show the correlation with these two agent benchmarks in the updated Figure 4. We also update Figure 1 to show the relations between general benchmarks with SWE-Bench/Tau2-Bench/Terminal Bench.
> The updated results also demonstrate that APTBench is a better proxy of the agentic ability of the pre-trained models.
>
> **Q4**: Typos.
>
> **A4**: Thank you very much for pointing this out. The typos have been fixed in the updated version of the paper.

---

> ### Comment · Area_Chair_gX5k · 2025-11-28
> **Reminder: Engage with Authors During Rebuttal**
>
> Quick reminder: the rebuttal period is still open, and the deadline is in less than one week. Please continue the discussion with the authors and share any clarifications or updates to your assessment before the rebuttal closes.

---

### Official Review · Reviewer_E5na · 2025-11-01

**Soundness:** 1
**Presentation:** 2
**Contribution:** 2
**Rating:** 2
**Confidence:** 3

**Summary:**

This paper introduces APTBench, a new benchmark designed to evaluate the "agentic potential" of base LLMs at the pre-training stage. The authors first argue that existing benchmarks (e.g., MMLU) correlate poorly with downstream agent performance. They then propose a new methodology: they define a taxonomy of core agentic skills (planning, action, and atomic abilities), convert multi-turn agent trajectories into single-turn multiple-choice and text-completion questions based on this taxonomy, and finally, validate the benchmark by demonstrating a strong correlation between its total score and performance on a downstream agent task (SWE-bench Verified).

**Strengths:**

Significant Problem & Interesting Entry Point: The paper correctly identifies a critical gap: the need to evaluate base models for agentic skills before the costly post-training stage. Its core technical idea—converting interactive trajectories into static, single-turn formats (MCQ/TC)—is a clever and pragmatic entry point to solving this challenging problem.

Useful Base Model Analysis: The experimental results, regardless of the benchmark's validation, provide valuable insights for the community. The observations on skill emergence with model scale (e.g., Qwen3-1.7B vs 4B) and the significant impact of agent-focused pre-training data are useful, independent contributions.

**Weaknesses:**

Fundamentally Unjustified Taxonomy and Thin Validation: This is the paper's primary methodological failure. For a paper whose topic is proposing a new benchmark, its main method and focus should be the rigorous selection and validation of what is being measured. This critical part is almost entirely omitted. The paper simply proposes a taxonomy ("Planning," "Action," "Atomic Abilities") based on intuition, with no formal analysis, theoretical grounding, or ablation studies to prove these metrics are necessary, sufficient, or comprehensive for measuring "agentic potential." Having skipped the step of validating its own metrics, the paper attempts to validate the entire benchmark by showing a holistic correlation with a single downstream task (SWE-bench). This validation is exceptionally thin and insufficient.

Potential for Benchmark Contamination & Weak Proxies: The benchmark's construction has other methodological issues. It is built from public datasets (SWE-Bench-Lite, GitHub repos) where the risk of data contamination for modern models is high and not deeply analyzed. Furthermore, the use of idealized README files as a proxy for a problem-solving trajectory in the EnvSetup task is a weak proxy that does not reflect real-world, feedback-driven interactions.

**Questions:**

The paper's central claim of measuring "agentic potential" relies on an intuitively chosen taxonomy (Planning, Action, Atomic Abilities) that lacks rigorous justification. Can the authors provide the empirical or theoretical evidence that these specific metrics are both necessary and sufficient? If not, should the paper's claim be significantly moderated to something more specific, such as "a static proxy for SWE-bench performance".

The benchmark's validation is based on a correlation with only one downstream task: SWE-bench. Why was this single, domain-specific task deemed sufficient? Were correlations against other agent domains (e.g., web browsing, research-focused agents) attempted? This single-task validation is not robust enough to support the paper's general claims.

The benchmark is constructed from public data (GitHub, SWE-Bench-Lite) that are likely present in the pre-training corpora of the models being tested. What steps were taken to quantify and mitigate this potential data contamination?

---

> ### Author Response · Authors · 2025-11-24
>
> Thank you for the careful review of the paper and valueable feedback. We will address all your concerns in the following points:
>
> **Q1**: Fundamentally Unjustified Taxonomy and Thin Validation.
>
> **A1**: **The fundamental core abilities of an agent are planning and action**, a view widely recognized in prior work [1,2]. An agent’s basic behavioral pattern is the Plan‑Action loop, and most other behavioral patterns are essentially extensions of this paradigm. To evaluate the agentic capabilities of base models, we therefore focus on these two dimensions: whether the model can decide the correct next step, and whether it can execute the step accurately through tool invocation.
> Currently, pre‑trained models lack a benchmark that explicitly defines and evaluates these abilities, so APTBench aims to provide such a standard. **Because base models generally lack strong instruction‑following capabilities, we adopt MCQ (multiple‑choice questions) and TC (text‑completion) formats** to evaluate these two abilities. This aligns with common practices in base‑model evaluation.
> The atomic abilities we include are additional dimensions tailored specifically to SWE and DR—two of the most widely discussed agent scenarios today. Since planning and action are inherently abstract, concrete scenarios are required to construct evaluation items; thus, we choose SWE and DR as grounding environments for measurement.
>
> **Q2**: Downstream task only has SWE-Bench.
>
> **A2**: We thank the reviewer to point it out. We have updated Figure 2 and Figure4 to show the correlations between APTBench with **SWE-Bench/Tau2-Bench/Terminal Bench**.
> We find that, **despite the differences in post‑training procedures, the final downstream results still show strong correlation with APTBench**, especially compared with other commonly used base‑model evaluation metrics. This is exactly what we aim for: APTBench can serve as a strong indicator of a base model’s agentic potential.
>
>
> **Q3**: About the data contamination issue.
>
> **A3**: For the EnvSetup tasks, we use the github repos of the research projects from the nearest NeurIPS/ICML/ICLR conferences, which are after many of the tested models knowledge cutoff. For other tasks, we also use newly generated trajectories that does not appears in the open web.
> To alliviate the contamination issue, in our paper we describe in detail the methodology for constructing the benchmark, so that new data can be used at any time to update APTBench. Other pre‑training teams can also use our approach to build their own in‑house evaluation sets for assessing the agentic capabilities of base models.
>
> [1] Yao S, Zhao J, Yu D, et al. React: Synergizing reasoning and acting in language models[C]//The eleventh international conference on learning representations. 2022.
>
> [2] Huang X, Liu W, Chen X, et al. Understanding the planning of LLM agents: A survey[J]. arXiv preprint arXiv:2402.02716, 2024.

---

> > ### Comment · Reviewer_E5na · 2025-11-26
> >
> > Thank you for the active response and the additional experiments.
> >
> > First, I appreciate the extra effort on the new experiments. Authors' newly added experiments resolves my concerns regarding Q2 and Q3.
> >
> > However, I'm still not convinced by the response on the Taxonomy (Q1). While I agree that planning and acting are important agentic capabilities, the issue is that you haven't demonstrated that this specific benchmark accurately measures those capabilities. Furthermore, you haven't proven that measuring these skills in base models guarantees they will be effectively inherited downstream. Conversely, good downstream performance does not necessarily prove that these capabilities were explicitly present or measurable during the pre-training stage. **The correlation you provided does not imply causation.** It is entirely possible that stronger models simply have the general capacity to perform well on both independent tasks (the benchmark and the downstream tasks). Therefore, I still believe the paper suffers from significant overclaiming.

---

### Meta-Review · Area_Chair_at1t · 2026-01-05

**Summary:**

This paper proposes APTBench, a benchmark intended to evaluate the “agentic potential” of base (pre-trained) LLMs, and aims to address the problem where (1) current pre-training benchmarks focus on static skills (e.g., knowledge or reasoning) rather than agentic capabilities, and (2) existing agent benchmarks are often end-to-end complex tasks and require the model to be able to follow multi-turn instructions and thus unsuitable for evaluating base models (which only go through pre-training). The core idea of transforming agent trajectories (from software engineering and deep-research tasks) into single-turn MCQ and text-completion questions is novel, practical, and well aligned with the constraints of evaluating base models.

Strengths:

1. All reviewers agree that the paper addresses an important and timely problem: how to assess agentic capabilities before post-training.

2. The methodology for converting multi-turn agent trajectories into lightweight, static evaluation items is novel and practically valuable.

3. The empirical analysis of base model scaling trends and the demonstrated correlation between APTBench and downstream agent benchmarks (SWE-Bench, Tau2-Bench, Terminal Bench; the latter two added during the discussion period) provides encouraging evidence that the benchmark captures signals beyond standard pre-training metrics.

Weaknesses (outstanding after the rebuttal):

A central concern remains unresolved across two reviewers (E5na and Cm9U), which is the proposed benchmark is not convincing for indicating the agentic capabilities. I agree with the reviewers’ concerns, after reading the paper, reviews and author response.

The paper’s taxonomy of agentic abilities (Planning, Action, Atomic Abilities) is insufficiently justified and validated. While the authors argue that planning and action reflect a widely accepted Plan-Act paradigm, Reviewer E5na still has concerns that (1) the benchmark does not convincingly demonstrate that its items actually measure these abstract capabilities rather than domain-specific knowledge. (2) Correlation with downstream performance does not establish that the measured capabilities are present during pre-training or are causally responsible for downstream agent success (there are many confounding factors in this validation approach). (3) Variance across domains and sub-tasks suggests possible entanglement with task-specific familiarity rather than general agentic reasoning. After the author's response, Reviewer Cm9U is still concerned with the mixed and sometimes difficult-to-interpret experimental results, without clearly interpretable absolute signals.

To summarize, I think both reviewers essentially question the validation of the usefulness of this benchmark, as the validation through current experiments and correlation analysis involves many confounding factors.

Overall Assessment:

The paper aims to address an important problem with a promising idea. However, in its current form, the work still falls short on rigorous validation and doesn’t show the proposed benchmark can fulfill the gap convincingly.

**Reviewer Concerns:**

The authors’ responses addressed several concerns during rebuttal:

1. They expanded correlation analysis beyond SWE-Bench to Tau2-Bench and Terminal Bench.

2. They clarified data generation procedures and alleviated the data contamination concerns.

3. They added LLM-as-a-judge results and supplementary NLL analyses for a model family.

 These updates made the experiments more comprehensive. The outstanding concerns/weaknesses are summarized above.

**Reviewer Scores:**

I think Reviewer Cm9U would have changed their score to 8. The other two reviewers already participated in the discussion and I don't think they would have changed their score even if they were allowed to engage more. Overall, I agree with the reviewers' outstanding concerns and think the paper could be more convincing after improving the rigor in validation.

---

### Decision · Program_Chairs · 2026-01-26

Reject